Subject Areas:
environmental science/environmental engineering/ecology

Keywords:
natural flood management, porous and non-porous hydraulic structures leaky barrier, acoustic Doppler velocimetry, fish behaviour, flooding

Author for correspondence:
Stephanie Müller
e-mail: MullerS1@Cardiff.ac.uk

# Leaky barriers: leaky enough for fish to pass?

Stephanie Müller[1], Catherine A. M. E. Wilson[1], Pablo Ouro[1,3] and Joanne Cable[2]

[1]Hydro-Environmental Research Center, School of Engineering, Cardiff University, Cardiff CF24 3AA, UK
[2]School of Biosciences, Cardiff University, Cardiff CF10 3AX, UK
[3]School of Mechanical, Aerospace and Civil Engineering, University of Manchester, Manchester M13 9PL, UK

SM, 0000-0002-3904-0400; CAMEW, 0000-0002-7128-590X;
PO, 0000-0001-6411-8241; JC, 0000-0002-8510-7055

Perceived as environmental-friendly hydraulic structures, leaky barriers used for natural flood management are introduced into rivers, potentially creating migration barriers for fish. Using sustainable, local materials to construct wooden barriers across river channels in upper catchments, these barriers aim to slow down the flow, reduce flood peaks and attenuate the flow reaching downstream communities. Yet little is known about their impact on hydrodynamics and fish passage. Here, we examined two model barrier designs under 100% and 80% bankfull flow conditions in an open channel flume. These barriers included a porous and a non-porous design, with the latter emulating the natural accumulation of brush, sediment and leaf material between logs over time. Flow visualization and velocity measurements recorded with acoustic Doppler velocimetry characterized the flow field upstream and downstream of the barriers. Our fish behavioural studies revealed that juvenile salmon (*Salmo salar*) movement between downstream and upstream sections of the flume was inhibited by barrier design rather than discharge, influencing upstream fish passage and their spatial preference, indicating the importance of barrier design criteria to facilitate fish movement.

## 1. Introduction

The introduction of artificial barriers worldwide has caused the removal, reduction, modification and fragmentation of the aquatic environment. This has resulted in habitat loss and degradation, and presents a major threat to fish worldwide and particularly in Europe [1,2]. In the UK, 99% of rivers are fragmented with only 1% of catchments free-flowing [3]. Low-head structures such as weirs, sluice gates, dams, culverts and water in- and out-takes, are used to divert flows, control and

**Figure 1.** (*a,b*) Leaky barriers installed into Wilde Brook, Shropshire, UK, typical of (*a*) porous and (*b*) non-porous structures found in the field (Photo credit: E. Follett). (*c,d*) Geometrically scaled (*c*) porous and (*d*) non-porous model structures constructed from horizontal wooden cylinders used in the current study; looking in upstream direction.

measure water levels, and ensure navigation, and hydroelectric facilities such as Archimedes screws and hydrokinetic turbines, are also responsible for 80% of the flow disruptions [3]. These structures can present physical and velocity barriers to fish movement and are, therefore, often equipped with fish passes. Failure to appropriately navigate the fish pass or hydraulic structure can inhibit fish movement and even delay migration [4]. The increased energy expenditure associated with the change in swimming gait needed to bypass these barriers may lead to premature fatigue and therefore, reduce the fish's chance to successfully reproduce [5]. In addition to these traditional structures, new hydraulic structures are being introduced into rivers, predominantly to mitigate the impact of flooding (e.g. leaky barriers [6]). Although these structures do not require fish to overcome a difference in head, they still alter the surrounding flow field, which may impact fish movement and habitat use.

Because new barriers are still being installed and not all traditional structures are obsolete and therefore cannot be removed, it is key to understand the interplay between physical properties and design, associated hydrodynamic alterations and their potential implications for fish movement. Hydraulic design of culverts used for road and rail crossings greatly impact on fish movement [7]. Due to the flow confinement and smooth surfaces created, fish are abruptly exposed to high streamwise velocities. Physical adaptations, such as an increase in channel roughness, however, generate secondary current cells, assisting smaller fish to overcome the barrier [7]. Sluice gates used to control and maintain water levels, on the other hand, have a backwater effect and increase upstream water levels. Depending on gate height, flow confinement may lead to overflow and high streamwise velocities beneath the structure and the formation of a recirculation zone or hydraulic jump. A study of an underflow sluice gate near a hydropower facility showed an increase in fish passage rate with increasing gate depth [5]. However, it was unclear whether this effect was caused by avoidance of the overflow or attraction to the higher velocities found beneath the gate [5]. Additionally, an increase in turbine passage was observed when the gate was lowered, potentially a result of the sluice overflow attracting fish towards the hydroelectric facility [5]. Similarly, hydraulic jumps observed downstream of weirs can distract fish from overcoming these barriers due to their attraction to turbulence [8].

The recently introduced nature-based flood management structures are often part of wider natural flood management campaign [6,9,10]. Due to the use of natural materials, these new structures, known as engineered leaky barriers, are perceived as environmental-friendly compared with traditional hard engineering flood management approaches. These barriers are constructed from wooden logs, fallen trees and branches (figure 1*a,b*). Installed into the upper catchment, they span the river channel cross-section. Due to their porous nature (figure 1*a*), these structures allow flow through them and, to facilitate unimpeded baseflow and fish movement, they are also designed with a vertical

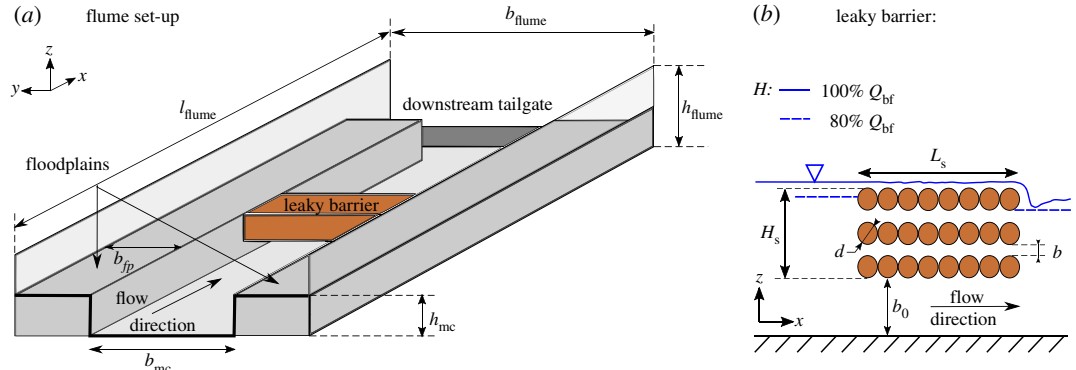

**Figure 2.** Schematic representing the experimental set-up showing a straight, rectangular compound channel of width $b_{mc} = 0.6$ m, bankfull height $h_{mc} = 0.15$ m and floodplains ($fp$) on either site of the main channel (mc) of width $b_{fp} = 0.3$ m. The porous and non-porous structure were placed within the main channel at approximately 5 m downstream of the flume inlet.

gap between the barrier and the river bed [6]. Under high flow conditions, the additional blockage created by these structures causes the flow to spill out onto the surrounding floodplains, making use of the floodplain storage. This enhances ground infiltration [11] to slows down the movement of both ground and surface water throughout the catchment and attenuates flow downstream [12]. Due to the natural accumulation of branches, sediment and leaf material between the log members over time, these structures can become more watertight, creating flows analogous to sluice and tidal gates.

Through laboratory experiments, we investigated the effect of idealized porous (figure 1c) and non-porous (figure 1d) model structures under two flow conditions, representing bankfull and near bankfull flow, on free fish movement and channel hydrodynamics. Both model structures were based on leaky barrier structures in Wilde Brook, Corvedale, Shropshire, UK, geometrically scaled by a factor of 0.15.

# 2. Material and methods

## 2.1. Flume set-up

Hydrodynamic and fish behaviour experiments were performed for an idealized porous and non-porous barrier in a recirculating open channel flume in the Hydro-Environmental Research Centre at Cardiff University, UK. The flume, presented in figure 2a, is 10 m long ($l_{flume}$), 1.2 m wide ($b_{flume}$) and 0.3 m deep ($h_{flume}$). While mean longitudinal flow was defined as positive longitudinal coordinate in $x$ direction, lateral and vertical coordinate were defined as $y$ and $z$, respectively. A longitudinal bed slope of 0.001 was applied to the flume. The flume comprises a straight compound channel with rectangular main channel cross-section of width $b_{mc} = 0.6$ m, bankfull depth $h_{mc} = 0.15$ m and floodplains of width $b_{fp} = 0.3$ m on either side of the channel.

A Sigmund Pulsometer pump (Sigmund Pulsometer Pump Ltd., type AM20A) controlled water discharge, and water surface elevation was adjusted by the tailgate weir located at the downstream end of the flume. Discharge ($Q$) and flow depth ($h_0$) remained fixed throughout the experiments. Prior to the installation of the barriers, uniform, subcritical flow conditions were established for bankfull ($Q_{bf}$) and 80% bankfull ($0.8Q_{bf}$) flow conditions, relating to a discharge of 0.028 and 0.022 m³ s⁻¹ and a flow depth of 0.15 and 0.13 m, respectively. These conditions represent the control treatment and a detailed breakdown is presented in table 1. Bankfull flow condition refers to the maximum discharge capacity of the main channel and therefore the greatest flow rate contained within the main channel before flow inundates onto the floodplains. Hence, 80% bankfull flow condition refers to 80% of maximum discharge capacity of the main channel and was selected to represent a higher probability of occurrence in one year than a larger magnitude bankfull event. The installation of the porous and non-porous structure resulted in a change in water surface profile generating gradually varied flow conditions. Flow depth was measured using a point gauge.

The bulk velocities and flow rates selected for the experiments are in the correct range to comply with Froude similarity. For Froude similarity, discharge and velocity scale use the following relationships: $U_{field} = U_{lab} = \sqrt{\lambda}$ and $Q_{field} = Q_{lab}\lambda^{5/2}$, respectively, where $\lambda = 6.7$. At Wilde Brook, we do not have a measurement of discharge at the leaky barrier locations and the selected lab discharges correspond to

**Table 1.** Details of the barrier structural characteristics and flow conditions upstream (1) and downstream (2) of the porous and non-porous structure, including cross-sectional blockage area ($A_s$), cross-sectional blockage ratio ($A_0$), solid volume fraction ($\phi$) and structural porosity ($\Phi$), mean flow depth ($H_1$ and $H_2$), difference between upstream and downstream mean flow depth ($\Delta H$, with * indicating overbank flow in column $H_1$), bulk velocity ($U_{01}$ and $U_{02}$), Reynolds number ($Re_1$ and $Re_2$) and Froude number ($Fr_1$ and $Fr_2$) based on hydraulic radius.

| | treatment | $A_s$ [m²] | $A_0$ [%] | $\phi$ [%] | $\Phi$ [%] | $H_1$ [mm] | $H_2$ [mm] | $\Delta H$ [mm] | $U_{01}$ [m s⁻¹] | $U_{02}$ | $Re_1$ [—] | $Re_2$ | $Fr_1$ [—] | $Fr_2$ |
|---|---|---|---|---|---|---|---|---|---|---|---|---|---|---|
| 100%$Q_{bf}$ | control | — | — | — | — | 146.3 | — | — | 0.32 | | 31 367 | | 0.32 | |
| | non-porous | 0.06 | 66.67 | 66.67 | — | 172.3* | 139.6 | 32.7 | 0.24 | 0.33 | 18 127 | 31 845 | 0.28 | 0.35 |
| | porous | 0.05 | 50.00 | 39.29 | 41.1 | 158.4* | 142.4 | 16.0 | 0.28 | 0.33 | 18 458 | 31 647 | 0.35 | 0.34 |
| 80%$Q_{bf}$ | control | — | — | — | — | 132.7 | — | — | 0.32 | | 25 424 | | 0.29 | |
| | non-porous | 0.06 | 66.67 | 66.67 | — | 157.5* | 124.2 | 33.3 | 0.22 | 0.30 | 14 521 | 25 931 | 0.28 | 0.32 |
| | porous | 0.05 | 50.00 | 39.29 | 41.1 | 140.5 | 131.0 | 9.5 | 0.26 | 0.28 | 13 852 | 25 522 | 0.36 | 0.3 |

field-scale discharges of 2.55 and 3.25 m³ s⁻¹, for the 0.8$Q_{bf}$ and $Q_{bf}$ conditions, respectively, which is in keeping with the field channel scale (bankfull flow area = 4 m²). The laboratory bulk velocity of 0.32 m s⁻¹ for bankfull conditions corresponds to a field-scale velocity of 0.85 m s⁻¹, which is reasonable for a stream of this magnitude.

## 2.2. Barrier structures

Both, porous and non-porous structure, were constructed using individual wooden dowels of diameter $d = 25$ mm, fixed in the main channel using silicon adhesive and located approximately 5 m downstream of the flume inlet, as shown in figure 2b. Each dowel row comprised eight dowels in longitudinal $x$ direction, with a barrier length ($L_s$) of 0.2 m and height ($H_s$) of 0.1 m. A vertical gap ($b_0$) of 50 mm remained between the lowest dowel edge and flume bed, designed to mimic field designs with similar cross-sectional flow blockage, allowing passage of baseflow and fish movement. A vertical distance $b$ of 12.5 mm was maintained between the dowels allowing flow through. Two barrier structure porosities were analysed by comparing a porous structure against a non-porous structure simulating the natural accumulation of sediment, leaf material and woody debris and therefore the clogging of the barrier. The latter structure was constructed by wrapping the external dowels in orange polythene to prevent flow through the structure while the 50 mm gap underneath the barrier structures remained present. Due to the presence of the vertical gap underneath both barriers, the channel cross-section is still deemed as porous. Here, we only concentrate on the porosity of the barrier structure itself.

River and barrier model designs was based on the geometric scaling of four length scales which characterize the physical properties of the stream and leaky barriers at Wilde Brook, Corvedale (Shropshire, UK) [13]. The model to prototype scale was approximately 1:7 (1:6.7) and based on geometric scaling of the (i) channel width, (ii) bankfull depth, (iii) vertical gap underneath a leaky barrier, and (iv) log diameter. For Wilde Brook the channel's $b_{mc}/h_{mc}$ ratio varies in the range $1.66 \leq b_{mc}/h_{mc} \leq 4.8$, based on 10 selected cross-sections and a set of 105 observations. The $b_{mc}/h_{mc}$ ratio of 4 was chosen for this study and previous studies on bed erosion [13] as this typifies the channel; this ratio was maintained in the lab model. At Wilde Brook the leaky barriers have a vertical gap to bankfull height ratio ($b_0/h_{mc}$) in the range of $0.333 \leq b_0/h_{mc} \leq 0.5$, which is typical of many leaky barriers in the field; a $b_0/h_{mc}$ of 0.333 was maintained for the laboratory model. The barrier model design used a dowel diameter, $d$, of 25 mm, which represents a typical field log diameter in the range $0.17 \leq d_{field} \leq 0.33$ m which is in keeping with the leaky barriers at Wilde Brook and other sites of this scale.

For each flow condition, a porous and non-porous structure were tested and one control condition, i.e. no structure present. Physical and hydraulic characteristics for each barrier are presented in table 1, including cross-sectional blockage area $A_s$, cross-sectional blockage ratio $A_0$, being the ratio between structural frontal area $A_s$ and flow area $A = H_{mc} \, b_{mc}$ as well as solid volume fraction $\phi = V_{structure}/V_{control}$, where $V_{structure}$ is the volume occupied by the solid barrier structure, defined as

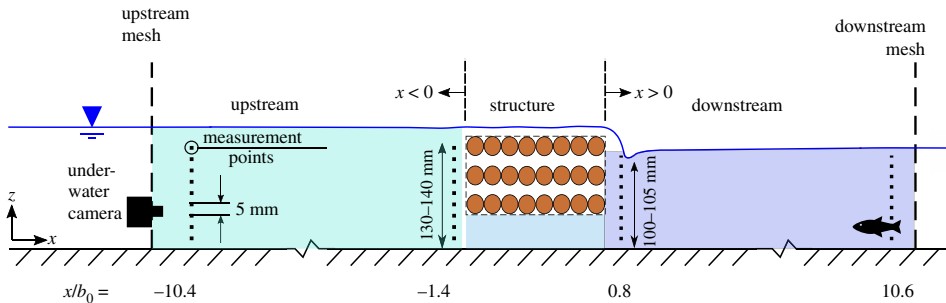

**Figure 3.** Schematic representing the position of the ADV measurements and fish behavioural trials test section. In total, four velocity profiles (upstream: $x/b_0 = -10.4$ and $-1.4$; downstream: $x/b_0 = 0.8$ and 10.6) were measured starting approximately 10 mm above flume bed until 30 mm beneath the water surface with a vertical resolution of 5 mm.

$\pi(d/2)^2 n_{\mathrm{dowels}} b_{\mathrm{mc}}$, and $V_{\mathrm{control}} = H_{\mathrm{mc}} L_s b_{\mathrm{mc}}$ is the volume for control conditions. The structural porosity of the porous design is defined as $\Phi = V_{\mathrm{pore}}/(V_{\mathrm{pore}} + V_{\mathrm{solid}})$, with $V_{\mathrm{pore}}$ the equivalent porosity volume resulting from the difference between $V_{\mathrm{control}}$ and $V_{\mathrm{structure}}$. Hydraulic characteristics are calculated for upstream (1) and downstream (2) locations, such as mean flow depth ($H_1$ and $H_2$), which was calculated from the average of the water surface measurements upstream and downstream of the structure. Additionally, the difference between upstream and downstream mean flow depths ($\Delta H = H_2 - H_1$) and bulk velocity ($U_{01}$ and $U_{02}$) were computed. Reynolds number was calculated as $Re = U_0 R_H/\nu$, with $\nu$ denoting the fluid kinematic viscosity and $R_H$ denoting the hydraulic radius. Froude number is $Fr = U_0/\sqrt{gR_H}$ with $g$ denoting gravity acceleration.

## 2.3. Velocity measurements and flow visualization

Flow velocities were measured upstream and downstream of the porous and non-porous structure using a sideways-looking acoustic Doppler velocimeter (ADV) (Nortek Vectrino) at a sampling rate of 200 Hz. To ensure sufficient data quality and to capture a representative sample of high-frequency turbulent fluctuations, measurements were conducted over 5–20 min. The water was seeded using Spherical® 110P8 hollow glass spheres (Potters Industries LLC) with a mean particle size of 11.7 μm and a specific gravity of 1.10 g cm$^{-3}$. During measurements, a signal-to-noise ratio (SNR) of at least 15dB and a minimum correlation of 70% were obtained by adding seeding material to enhance signal quality. Velocity profiles were taken at two locations upstream ($x/b_0 = -10.4$ and $-1.4$) and two downstream ($x/b_0 = 0.8$ and 10.6) for the two structures and for both flow conditions, as indicated in figure 3. For the control treatment, velocity measurements were conducted over a single vertical profile at $x/b_0 = 0.8$. The velocity profiles consisted of 20–26 point measurements with a vertical resolution of 5 mm. Velocity data were filtered and post-processed using Matlab (R2018b and R2019b). In a first pre-filtering step, data with insufficient SNR and correlation were removed before the data were despiked using an open-source despiking filter of Mori *et al.* [14,15], which is a modification of the three-dimensional phase space threshold filter by Wahl [16]. Time-averaged velocities were calculated from the time history at every measured point and denoted by an overbar. Additionally, flow patterns of the near wake were visualized for both structures and flow conditions using Flourescent Fwt red (Cole-Parmer Instrument Company Ltd), which was injected for flow visualization at the centreline upstream of the structure at a range of elevations. A GoPro Hero 5 underwater camera (1920 × 1080 px, linear mode) positioned at the left-hand side of the main channel recorded the dispersion of the dye tracer.

## 2.4. Fish passage experiment

Fish passage behaviour tests were conducted between 21 January and 1 February 2019 between 7.30 and 22.00. Atlantic salmon (*Salmo salar*, mean ± s.d. mass 12.3 ± 5.4 g, mean ± s.d. standard length 93.3 ± 13.6 mm), sourced from Kielder Salmon Centre were maintained within the Cardiff University Aquarium at 14 ± 1°C. For the experiment, juvenile salmon standard length correspond to the length of adult salmon (500 ≤ standard length ≤ 800, [17]) so this aspect complies with geometric similarity.

In general, Atlantic salmon are anadromous, they hatch in rivers and streams, migrate to the sea to mature, and then migrate back to their birthplace to spawn. Growing up in fast flowing, clean and well oxygenated freshwater, these fish demonstrate positive rheotaxis, orientating themselves against the flow direction (Kalleberg (1958) in [18]). It should be noted that at the time of the experiment, fish were at their parr-early smolt stage and therefore did not show migratory behaviour. This study focuses on the free movement of these fish in the vicinity of a porous and non-porous structure. Swimming tests, conducted by Palstra *et al.* [19], with juveniles ($29.9 \pm 0.9$ g, standard length $123 \pm 16$ mm) of similar length scale to our model fish, determined a critical swimming speed at $0.959 \pm 0.103$ m s$^{-1}$; however, it should be noted that swimming performance is strongly dependent upon fish size [19].

The experimental test section, shown in figure 3, was 1.6 m long and bounded by a plastic mesh with square holes of size $5 \times 5$ mm, spanning the entire width of the flume, at approximately 0.7 m ($14b_0$) upstream and downstream of the structure. For convenience in the analysis, the test section was divided into three spatial zones: upstream ($\Delta x = 0.7$ m), downstream ($\Delta x = 0.7$ m) and structure ($\Delta x = 0.2$ m). A GoPro Hero 5 underwater camera was positioned $x = 0.7$ m upstream of the barrier, on the main channel centreline outside the test section and pointed in the downstream direction. The water was dechlorinated using Seachem Prime Concentrated Conditioner and chilled to 14°C.

Each fish was gently transferred to the flume and given a 15 min acclimatization period including a 2 min incremental increase in discharge over the first 10 min up to the test discharge level, followed by a 5 min acclimatization at this test discharge (22 or 28 l s$^{-1}$) before each trial commenced ($n = 16$ for porous, 80% bankfull discharge; otherwise $n = 14$ per treatment). Each trial lasted 10 min where individual fish were released at the most downstream end of the test section along the centreline of the main channel. Treatment order could not be randomized because of the construction method of the installed barriers. The porous structure was tested first, followed by non-porous structure and control condition. For each treatment, 80% bankfull discharge was tested prior to 100% bankfull flow condition. Human intervention took place only in cases where fish remained stationary in the furthest downstream transect and refused to swim. In this case, fish were gently nudged with the handle of the net, and in the 35 cases this occurred this stimulus worked for 26 of the fish. The nine non-responding fish were excluded from the analysis (included in analyses: $n = 10$ for control 80%$Q_{bf}$, $n = 12$ for control 100%$Q_{bf}$, $n = 13$ for porous 80%$Q_{bf}$ otherwise $n = 14$). During each test, time spent in each zone (upstream, downstream or underneath barrier) and number of upstream passes as a measure of movement activity was recorded manually using stopwatches and the underwater camera.

Statistical analysis was conducted using R v. 3.6.3 statistical software. Spatial preference was analysed using a separate general linear model (GLM) with Gaussian distribution and identity link for each spatial zone, allowing the investigation of the difference in mean between time spent upstream, downstream and underneath the structure (time proportion as dependent variable) and barrier treatment as well as flow condition (independent variables). Association between number of upstream passes per fish (dependent variable) and leaky barrier as well as flow condition (independent variable) was tested using a GLM with Poisson distribution and identity as link function. A binomial GLM with logit link function was conducted to analyse potential associations between flow condition as well as barrier and upstream passed fish and flood plain usage which are reported as categorical variables (passed/not passed or used/not used). Non-significant variables were stepwise removed from the statistical analysis and residuals were used to assess the suitability of the tests. *p*-value significance was taken at 0.05.

# 3. Results

Channel hydrodynamics and fish behaviour were quantified for a porous and non-porous barrier and a control situation (no barrier) and two flow conditions to determine the impact of flow conditions and structure on spatial preference and upstream passage of juvenile Atlantic salmon.

## 3.1. Hydrodynamics

Normalized time-averaged longitudinal velocity ($\overline{u}/U_{01}$) results for the main channel hydrodynamics measured for the control situation at $x/b_0 = 0.8$ (*a*), and for the non-porous (*b*) and porous barrier (*c*) profiles for 100% (green) and 80% bankfull (blue) flow conditions at the four $x/b_0$ locations indicated in figure 3 are presented in figure 4. The measured velocity profiles for the control scenario represent typical open-channel flow conditions following a logarithmic distribution, with slightly higher normalized mean longitudinal velocities found for 80% bankfull discharge, indicating higher

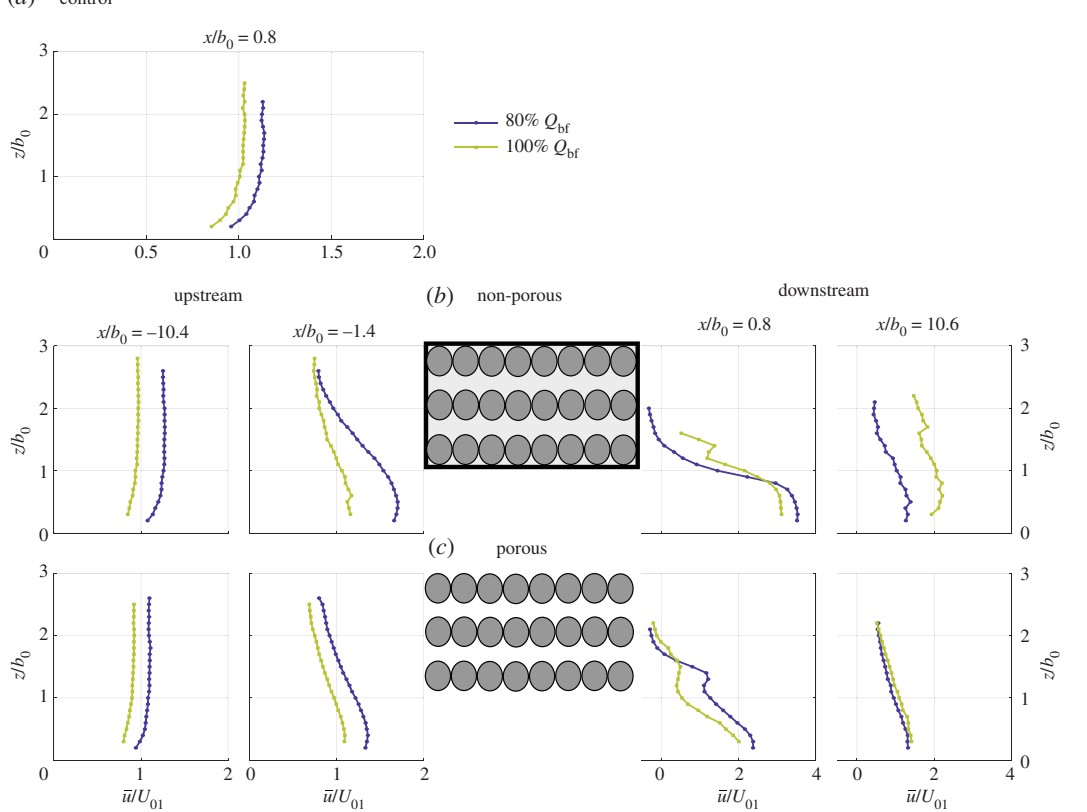

**Figure 4.** Mean longitudinal velocity profiles obtained under 80% (blue) and 100% (green) bankfull flow condition for control (*a*), non-porous (*b*) and porous (*c*) barriers.

distribution of momentum in the centre of the channel (figure 4*a*). This is caused by sidewall effects and the associated hydrodynamics redistributing momentum towards the channel centre under the 80% bankfull discharge.

Upstream velocity profiles for the non-porous and porous barriers are shown on the left-hand side of figure 4*b*,*c*, respectively. Furthest upstream ($x/b_0 = -10.4$) of the barriers, velocity profiles still follow a near-logarithmic distribution similar to that recorded under control conditions. Immediately upstream of the structure ($x/b_0 = -1.4$) higher values of $\overline{u}/U_{01}$ occurred for the 80% bankfull flow conditions regardless of the leaky barrier structure. For the non-porous design, however, there is higher momentum flow going through the bottom gap, changing the velocity distribution with the highest values occurring at mid-gap height ($0.5b_0$) and progressively decreasing towards the water surface. Such changes in the longitudinal velocity distribution for 80% bankfull flow conditions are more subtle for the porous barrier as a result of a decreased flow blockage as this structure allows through flow. For the non-porous case, due to its blocking-nature (figure 4*b*) there is a 20% increase in upstream flow depth for both flow conditions compared with control conditions (see table 1). This resulted in overbank flows for both discharges, which led to lower in-channel mean longitudinal velocities upstream of the barriers. For the non-porous structure the main channel flow depth exceeded bankfull flow depth by 15% and 5% for bankfull and 80% bankfull discharges, respectively, while for the porous structure this only increased by 8% and 6% for bankfull and 80% bankfull discharge, respectively, with overbank flow only observed for bankfull flow conditions (table 1).

Downstream velocity profiles are shown on the right-hand side of figure 4*b*,*c* for the non-porous and porous barriers, respectively. Immediately downstream of the non-porous leaky barrier ($x/b_0 = 0.8$), the maximum $\overline{u}/U_{01}$ is found at approximately one third of the gap height ($0.33b_0$) and increased 2.7 and 2.0 times compared with values at $x/b_0 = -1.4$ (figure 4*b*) for 100% and 80% bankfull flow, respectively. The maximum $\overline{u}/U_{01}$ was 10% higher for the 80% bankfull discharge than for bankfull conditions, as in the latter case, the upstream flow spills onto the floodplains and overtops the barrier, redistributing momentum from the main channel and more specifically, from the 'under flow' region beneath the structure. In all cases, velocity profiles show a progressive decrease in $\overline{u}/U_{01}$ with increasing elevation

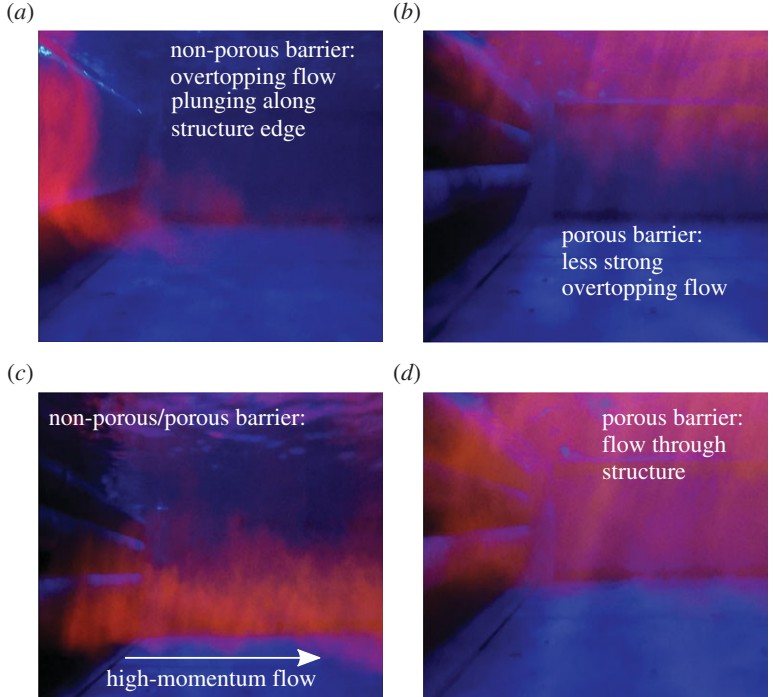

**Figure 5.** Near wake flow visualization of both barriers (barrier on the left-hand side) was conducted using fluorescent dye, injected at different heights upstream of the barrier. Flow alterations observed include overtopping flow (non-porous (a) and porous (b)), high-momentum flow (c) and flow though the porous barrier (d). The fluorescent dye is indicated in red with flow direction being left to right.

in the water column. Immediately downstream of the porous barrier, the maximum longitudinal velocity only increased by approximately 1.8 times compared with $x/b_0 = -1.4$ for both flow conditions (figure 4c). Longitudinal mean velocities were still slightly higher for 80% bankfull flow, probably due to the overbank flow observed under bankfull conditions, as well as the increased flow through the barrier. A notable feature in the wake of the leaky barrier is a second peak featuring a slight increase in longitudinal mean velocity at the lowest inter-dowel gap, i.e. $z/b_0 = 1.5$, as a result of flow going through the porous barrier. With increasing downstream distance, longitudinal velocities start to recover. Far downstream, at $x/b_0 = 10.6$, the difference in $\overline{u}/U_{01}$ between discharges was more pronounced for the non-porous barrier with higher longitudinal mean velocities under the bankfull flow. By contrast, velocity recovery was found to be independent of discharge for the porous barrier, probably due to the reduced overtopping flow.

As the pointwise measured velocity profiles only provide information about the time-averaged velocity statistics, flow visualization was also used to observe the instantaneous flow field in the wake of the porous and non-porous barriers. Dependent on the barrier, the downstream flow field was characterized by high-momentum flow, through flow and overtopping flow which can be seen in figure 5. The significant difference in overtopping flow observed for both barriers is shown in figure 5a,b for 100% bankfull flow. In particular, in the case of the non-porous barrier, the overtopping flow was observed to plunge along the barriers' edge, joining the high-momentum flow from the under-flow region (figure 5c). Less strong overtopping flow was present for the porous leaky barrier due to the flow through the barrier (figure 5d). No significant overtopping flow was observed at 80% bankfull flow conditions for both barriers.

## 3.2. Fish behaviour

Fish behaviour results were analysed in terms of time fish spent downstream, upstream and underneath the barrier as well as percentage of fish passing from the downstream into the upstream region, and mean number of upstream passes per fish (figure 6). While no significant changes between 80% and 100% bankfull flow were observed for spatial preference and upstream fish passes, the barrier design did impact fish behaviour.

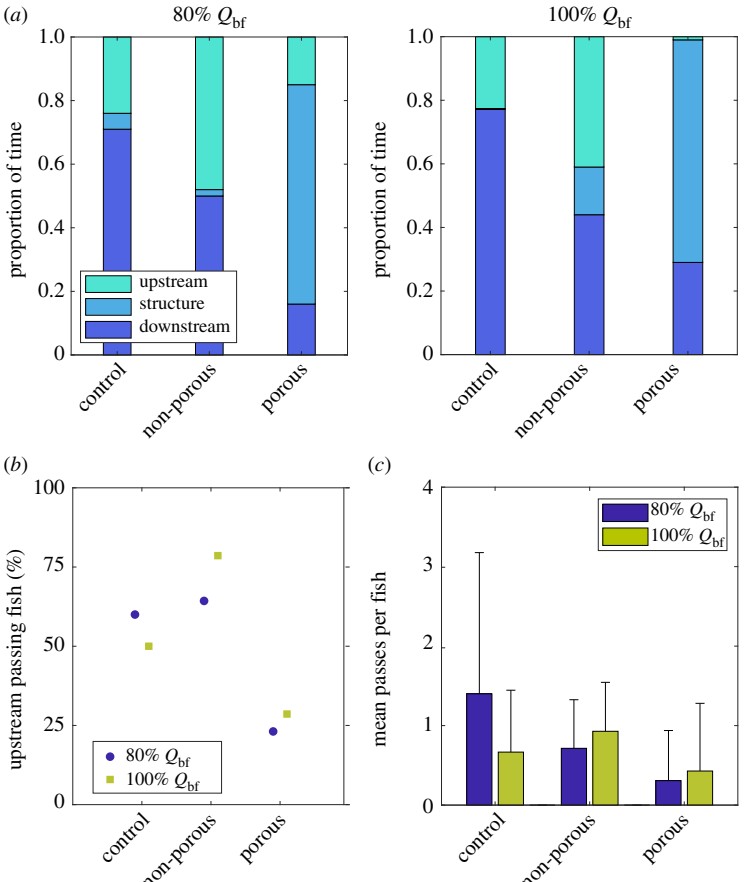

**Figure 6.** Summary of fish behavioural test showing (*a*) average time fish spent downstream (blue), beneath the structure (light blue) and upstream (green) under control, non-porous and porous barrier for 100% (right) and 80% (left) bankfull flow conditions. Percentage of fish passing from downstream area into upstream area is presented in (*b*) for 80% (blue) and 100% (green) bankfull flow conditions. Mean upstream passes per fish are shown in (*c*) with error bars representing standard deviation.

Mean proportion of time fish spent downstream, upstream and underneath the barrier after being released are shown in figure 6*a* for 80% bankfull (left) and 100% bankfull (right) flow conditions. Spatial preference was significantly more impacted by barrier porosity (GLM, all $p < 0.001$) than by an increase in discharge (GLM, all $p > 0.001$). Independent of the flow condition, fish spent more time downstream when no leaky barrier (control) was placed inside the test section (71% and 77% for 80% bankfull and bankfull discharge, respectively). Time spent downstream significantly differed between control condition and barriers present (GLM, porous: $p < 0.001$ and non-porous: $p = 0.0057$), but also among barriers (GLM, $p = 0.0097$). By contrast, time spent upstream only significantly differed from the control case when the non-porous barrier was present (GLM, $p = 0.0239$). In case of the non-porous leaky barrier, fish spent similar time upstream (48% and 41%) and downstream (50% and 44%) for 80% and 100% bankfull flow condition, respectively. An increase in time spent in the upstream section of 100% and 78% compared with control condition was observed for 80% and 100% bankfull flow, respectively. Time spent underneath the non-porous barrier increased from 2% to 15% when increasing the discharge but did not differ significantly from the control condition (GLM, $p = 0.341$). Conversely, in the presence of the porous barrier, fish spent most time underneath the barrier, demonstrated by 69% and 70% for 80% bankfull and 100% bankfull discharge, respectively, which significantly differed from the control condition (GLM, $p < 0.001$) as well as from what was observed for the non-porous barrier (GLM, $p < 0.001$). Similar to what was observed for the non-porous barrier, fish spent equal time upstream (15%) and downstream (16%) under 80% bankfull flow condition; however, under 100% bankfull flow time spent downstream increased to 29% while time spent upstream decreased to 1%.

The percentage of fish passing at least once from the downstream into the upstream region for 80% (blue) and 100% bankfull (green) flow conditions is presented in figure 6*b*. No significant association was

found between flow condition and percentage of upstream passing fish (GLM, $p = 0.9667$), however, a significant difference between control situation and the presence of the barriers was noted (GLM, $p = 0.0026$). While in the absence of a barrier (control), 60% and 50% of the tested fish passed at least once into the upstream region under 80% and 100% bankfull flow, respectively, a higher percentage of fish passed upstream when a non-porous barrier was present (GLM, $p = 0.0665$) and smaller percentage of fish passed upstream under the presence of the porous barrier (GLM, $p = 0.1573$). The percentage of fish passing upstream was significantly different between the porous and non-porous barrier (GLM, $p = 0.0012$).

As every fish was able to pass multiple times from the downstream region into the upstream region, figure 6c presents the mean number of passes per fish. Despite no significant association between number of upstream passes per fish and flow condition (GLM, $p = 0.9963$), the mean number of upstream passes per fish significantly differed when a barrier was present (GLM, $p = 0.0174$). Under the control treatment, every fish passed on average 1.4 and 0.67 times from downstream to upstream under 80% and 100% bankfull discharge, respectively. This number decreased not significantly in the presence of a non-porous leaky barrier (0.71 and 0.93 times for 80% and 100% bankfull discharge; GLM, $p = 0.5138$), but even more so when a porous barrier was present which led to a significant difference to the control condition (0.31 and 0.43 times for 80% and 100% bankfull discharge; GLM, $p = 0.0096$). In addition, a significant difference in mean passes per fish was found among both barriers (GLM, $p = 0.0297$). Highest variation in mean passes per fish was found for the control treatment.

When increasing the discharge to bankfull flow or under blockage (i.e. non-porous structure) conditions, the upstream water level rose, inundating both floodplains on either side of the main channel and therefore, opening potential, new habitat. Interestingly, under these conditions, a small, not significant, minority of fish (7% for non-porous barrier, both discharges and 17% for control, 100% bankfull discharge) used the additional space by swimming onto the floodplains. Hence, no significant association was found between floodplain usage and discharge (GLM, $p = 0.3304$) and leaky barriers presence (GLM, $p = 0.1621$).

To analyse the impact of human interaction on fish behaviour, all tests were performed with and without nudged fish. Similar significant independent variables were calculated for the percentage of upstream passing fish, floodplain usage and time spent upstream, downstream and beneath the barrier. Only the dependent variable 'passes per fish' resulted in a different result when prodded fish were excluded, showing that neither flow condition (GLM, $p = 0.8893$) nor barrier (GLM, $p = 0.119$) had a significant impact. In comparison, barrier was found to be significant factor influencing fish behaviour (GLM, $p = 0.0174$) when nudged fish were included.

## 4. Discussion

We have shown that our porous and non-porous barriers with a vertical gap did not prevent fish movement, but they did impact fish behaviour. Interestingly, the difference in discharge was not the decisive component impacting fish behaviour, instead the physical design of the barrier and the structure's porosity was more important. Spatial preference, percentage of upstream passaging fish and passage number varied among the tested barriers which may be linked to a range of reasons including for example hydrodynamic alterations of the surrounding flow field, visual cues or the provision of shelter.

Under high flow conditions, both barriers aim to reconnect the main channel flow zone with the adjacent floodplain zone. By doing so, water backs up and spills onto the floodplains and inundates them, creating new habitat for aquatic organisms but also supporting upstream nutrient and sediment exchange [20]. Floodplains often contain wood in the form of logs, trees, branches and brush with high densities of macro-invertebrates and therefore potentially provide additional food sources for fish [21], but also spawning and nursing grounds during high flow periods [20,22]. However, not all flow conditions led to floodplain inundation. It strongly depends on discharge and channel cross-section as well as physical properties of the barrier. Although in our experiment fish explored the floodplain regions characterized by low velocities, this may not be the case in the field with natural increased boundary roughness and predators.

When comparing 100% and 80% bankfull flow conditions, no differences in fish behaviour were observed, probably due to the fact that velocity magnitudes did not vary greatly between the analysed flow conditions. In addition, salmon, whether juvenile or adult, are relatively strong

swimmers, well adapted to high flow velocities. However, the size of fish size used in our experiment might not be appropriately scaled to the experimental flow conditions, a limitation of the current study. A higher increase in absolute longitudinal mean velocities was observed for the lower discharge, less impacted by overbank flow, which mostly occurred at higher discharge and barrier blockage. These barriers not only impact upstream and downstream flow depth, they strongly alter the velocity distribution in its vicinity. A range of flow variations were detected in our experiments including overtopping flow, flow through the porous barrier, and high-momentum flow from underneath the barrier. These high longitudinal velocities beneath the barrier prevent this gap, and therefore the channel cross-section, from being blocked by sediment, leaf material and woody debris accumulation when installed in the field but did not prevent fish from moving into the upstream region, indeed fish actually sought shelter underneath. For the non-porous barrier, in particular, a larger number of fish passed upstream and spent more time in this flow volume, despite the slightly higher longitudinal mean velocities. This may indicate that higher momentum flow provides a clearer cue for fish of where to pass. Such flow acceleration may not only impact fish behaviour but also alter the river bed by creating scour and by promoting sediment transport and generating scour, potentially leading to the exposure or smothering of eggs at spawning grounds as well as survival of benthic macro-invertebrates [12,23]. Complex, wooden barrier arrangements have the ability to create complex habitats with riffles and pools, fostering an increase in fish abundance, species richness [12,24] and biomass (e.g. for largemouth bass [25]).

Besides the flow alterations caused by the non-porous and porous barriers, physical appearance may impact the spatial preference and passage of fish. In the current study, the main physical difference between the tested barriers was the orange polythene wrapping, which prevents through flow, but also created a more unnatural, coloured obstacle compared with the porous structure which clearly shows natural elements in the form of wooden dowels. Depending on species, fish are able to differentiate colours and are attracted to different colours [26]. For instance, while bluegill sunfish and young carp react more towards red [26,27], Japanese marine fish species show greater preference for blues and greens [28]. Salmonids do possess colour vision (e.g. masu salmon) [29], but little is known about their attraction to colours. Thus, the coloured wrapping of the non-porous leaky barrier might have acted as a visual cue, guiding fish upstream. In this context, it should be noted that fish perception of the barrier colour may have been compromised due to variations in ambient light.

The natural appearance of the porous barrier may increase the attractiveness of this structure as a fish shelter, potentially being the reason why fish spent more time underneath the barrier and therefore passed less often upstream. Overhanging logs and complex accumulations of wood, or as in our case wooden cylinders, are an important source of cover in rivers, which provides habitats for different species [12]. For instance, strong preference for overhead cover has been reported for Atlantic salmon at lower temperatures [30]. Besides cover, complex wooden structures are an important refuge for small fish against predators by causing visual interference and entry prevention [31]. When comparing naturally occurring complex wooden structures against installed wooden structures, largemouth bass selected both structures at a similar rate [32].

In general, our porous and non-porous barriers do not present a barrier to fish movement if certain design criteria are fulfilled, such as provision of a gap underneath the structure allowing unimpeded base flow and fish passage [6]. Therefore, regular maintenance of these barriers is required in the field to prevent further blockage by driftwood, debris, sediment, leaf material or inorganic materials, leading to the creation of a physical, solid barrier to fish movement. Such field monitoring may also increase the lifespan of the barrier, but adds to the cost. Alternatively, the public could be engaged to document barrier states by submitting photographs and/or being involved in community conservation projects to clear barriers of debris after a flood event. In the meantime, further research will be needed to assess blockage and structural decay over time.

Generalization of the current results is limited as the study was only performed on one species of a particular size category, under strong lighting conditions in a simplified, scaled environment. In their natural environment, fish at different life stages will show considerable variation in response dependant on, for example, past experience, noise, predators, feeding and hydrodynamics [33]. Although the transferability of our findings from the porous and non-porous model structures into the field is limited due to the unrealistic scale of vertical gap to fish and the simplified design of the barriers, a range of potential ecological and hydrological advantages were noted. Together with further work, our findings may be of relevance in the design of instream structures such as leaky barriers used for natural flood management. So far, design guideline is limited and only a few studies have assessed the impact of leaky barriers on fish movement.

# 5. Conclusion

The impact of a porous and non-porous structure on channel hydrodynamics and fish behaviour (*Salmo salar*) were experimentally investigated for two flow conditions. We show that barrier porosity, rather than discharge, was the decisive component impacting fish movement and spatial preference. Fish movement was influenced by porosity, with more fish undergoing upstream passage for the non-porous design compared with its porous counterpart. This highlights the importance of barrier porosity as design parameter for hydraulic structures. This study, together with further research, may play an important role in the design and delivery of engineered leaky barriers as green, eco-friendly hydraulic structures used for natural flood management while ensuring the mitigation of flooding, maintaining habitat and enhancing connectivity for aquatic organisms.

Ethics. All fish behavioural experiments were approved by Cardiff University Animal Ethics Committee and conducted under Home Office Licence PPL 303424 following the ARRIVE guidelines [34].

Data accessibility. Data underpinning the results presented here are available in the electronic supplementary material and can be found in the Cardiff University data catalogue at http://doi.org/10.17035/d.2021.0129240456.

Authors' contributions. Concept and study design by C.A.M.E.W., J.C. and S.M. Experiments were conducted, and manuscript drafted by S.M. Hydrodynamic data analyses and interpretation by S.M, P.O. and C.A.M.E.W., with J.C. and S.M. assessing fish behaviour. All authors edited and revised the manuscript critically and approved the submitted version.

Competing interests. We declare we have no competing interests.

Funding. This research was funded as part of the Water Informatics Science and Engineering Centre for Doctoral Training (WISE CDT) [EP/L016214/1] from the Engineering and Physical Science Research Council (EPSRC).

Acknowledgements. We thank Paul Leach, Steven Rankmore, Gareth Castle, Valentine Muhawenimana and Jelena Nefjodova for technical assistance, Rhi Hunt for providing statistical advice and the reviewers for their valuable comments.

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
