## [Peer Review File · Royal Society Open Science]

Review History

RSOS-201843.R0 (Original submission)

Review form: Reviewer 1

Is the manuscript scientifically sound in its present form?

Yes

Are the interpretations and conclusions justified by the results?

No

Is the language acceptable?

Yes

Do you have any ethical concerns with this paper?

No

Have you any concerns about statistical analyses in this paper?

No

Recommendation?

Major revision is needed (please make suggestions in comments)

Comments to the Author(s)

The experiments are interesting and it is challenging to perform hydraulic experiments with living organisms. I do think there is a publishable study in here but, in its current form I have concerns detailed below. The main one is that, to me at least, these experiments are highly abstract and not comparable to conditions in 'field scale' rivers – the scaling in these experiments does not appear to be meaningful for the fish, leaky dam or the hydraulics. Therefore, I have concerns about the relevance of the findings to leaky dams and fish in rivers. However, I think this is made into a bigger issue than it needs to be due to the pitching of the whole paper around leaky barriers. I would personally rework to focus more around fish behaviour in experimental conditions and how fish navigate around and through structures, and then have a much more brief mention at the end of the discussion that these results, with further work, may be of relevance for x,y,z reasons, one of which may very well be leaky dam design. I hope the comments below help the authors.

Page 1, Line 10 beginning "If global warming..." – should the reference be at the end of the sentence? Presumably the 220% is a prediction by the paper referenced earlier in this sentence.

Page 2, Line 25: there is a typo in this sentence.

Page 2, Line 38 beginning "The direction..." reads strangely to me.

Page 2, Line 46 - 50: I found this section a bit confusing. I understand it is a scale model but scaled to what? – I am guessing this isn't a Reynolds or Froude scaling approach but, some information on how the flow is scaled relative to barriers is needed if results are going to be used to inform field-scale work. In addition, are these flows scaled to a particular river? The discharge seems low relative to the depth, meaning the velocity is high (presumably around 0.3 m/s) – I cannot see how these conditions could be 'scaled up' to be relevant to field-scale river conditions.

I also wanted to see a breakdown of control flow conditions at this point but I noticed it came later. Might be worth referring to that table here too.

Page 2, Line 54: Given the condition is 80% bankfull it might be best to describe it as a "higher return period" than as a "higher flood return periods"

Page 3 – Figure 1 is useful but I still have difficulty visualising the barriers. A photograph of the barriers would be a useful addition.

Page 3, Line 30 - 44: "scaled from those installed in the field". As above, more information about this scaling is needed – what measurements of the field barriers were made and how was the scaling done? For example, does the 12.5 mm gap scale to any field measure? The manuscript states that two leaky barrier porosities were analysed but no information is given on where these come from and whether they are based on field measurements. I suspect these are completely abstract structures which is not a problem in itself but, I do think a refocusing of the manuscript away from 'recreating leaky barriers in rivers' is probably needed.

Page 5, line 51: Maybe I have misunderstood but the water depth is 15 cm yet the fish are approximately 12 cm long. Again, the scaling here seems way off to me – especially if this is meant to be bankfull flow. Given it is known the ability of salmon to overcome and navigate barriers is dependent on the ration of upstream and downstream water depth this strikes me as a problem. Plus, the 'gap' under the barrier was 5 cm high which again, doesn't strike me as a relevant fish-structure scaling for a real world situation. As above, this is much more of a problem if the purpose is to recreate river conditions, which appears to be the aim. It is much less

of a problem if the focus is on fish behaviour more generally around porous and non-porous structures.

Page 7, line 54: I think it would be useful to define the 3 regions, particularly “under flow” – presumably this means the flow beneath the structure in the opening but confirmation of this would be good.

Page 7, line 55 beginning “In all cases...” this sentence is worded oddly to me.

Page 8, line 53: typo and strange characters in this sentence.

Page 10 – Figure 5 – I got really confused here but, I think the colours are mislabelled in the caption. Downstream is dark blue not green, etc. Also, should figure 5c be colour coded?

Page 11 line 38 – 45: I agree with this summary of the findings, which are interesting in informing about how fish interact with an abstract obstacle but I just do not see given the scaling of fish to barrier, and the depth to velocity scaling of the flow, how this can really be related to fish behaviour around field-scale, leaky barriers.

Page 12, line 27: What colour was the polythene wrapping?

Page 12, line 35: I don't find the colour argument that convincing. Through rheotaxis the fish will know which way is upstream. I do agree that the colour may influence their behaviour.

Page 12, line 40: There is a transition here from discussion about colour to overhanging logs. I would separate into 2 separate paragraphs

Page 12, line 46-57: I agree but none of this content is really informed by the experimental findings.

Page 13, line 8 – 12: I agree with all this and I think this is an interesting finding. The second half when referring to leaky barriers I find more problematic because there are so many confounding influences and because the results here are far from conclusive.

Review form: Reviewer 2

Is the manuscript scientifically sound in its present form?

No

Are the interpretations and conclusions justified by the results?

Yes

Is the language acceptable?

Yes

Do you have any ethical concerns with this paper?

Yes

Have you any concerns about statistical analyses in this paper?

Yes

Recommendation?

Major revision is needed (please make suggestions in comments)

Comments to the Author(s)

This paper presents an experimental study of flow fields and fish behaviour around leaky wooden dams of two different porosities and at two different discharges. Leaky wooden dams are increasingly being used in small headwater streams to contribute to "slowing the flow". I have concerns over a number of aspects of this study:

1. scaling of channel and wood dimensions (scaling is not adequately presented; are you Froude scaling or merely applying geometric similitude? We need more information about the field prototype to know)
2. scaling of fish relative to the above (how does the width and height of your model salmon compare to prototype salmon? How does this scale compare to your channel scale (width, depth)? How do either of these scales compare against your wood structure scale (both the elements and their spacing)? How do all these compare against the scale of the burst and/or sustained swim speed of your model salmon to prototype salmon?)
3. fish behaviour- generally speaking, a juvenile salmon doesn't really want to go up or downstream and it would be perfectly fine/happy if it never passed your barrier. If it was an anadromous salmonid (the next stage in the life cycle), it would swim seaward when the time was right, but given their size I don't think that would happen for another year. So the question is the extent to which "motivation" was dominated by the fright response of the fish to being prodded and whether fish were hiding underneath the porous barrier and exhibiting avoidance behaviour when they swam upstream? Unfortunately, we cannot tell. I think that all prodded fish must be removed from your analyses in order to eliminate these possibilities.
4. the real-world applicability of leaky wooden dams preventing access to headwaters by adult salmon; leaky wooden dams are generally installed in headwater streams that are not reachable by adult salmon, so it is unclear how applicable your results are in the real world.

In addition to the above, i think that this study would benefit from inclusion of additional velocity data and exploration of the jet recovery downstream of structures. I have seen this data presented in non peer-reviewed outlets by your team, so it should not be too challenging to include it here; i think that the contrast between hydraulics of porous/non-porous barriers is of significant interest.

I have made some other edits and suggestions on the manuscript for your consideration (see Appendix A).

Decision letter (RSOS-201843.R0)

This year has been very difficult for everyone, and we want to take the opportunity to thank you for your continued support in 2020.

The Royal Society Open Science editorial office will be closed from the evening of Friday 18 December 2020 until Monday 4 January 2021. We will not be responding during this time. If you have received a deadline within this time period, please contact us as soon as possible to allow us to extend the deadline. If you receive any automated messages during this time asking you to meet a deadline, we offer apologies and invite you to respond after the festive period or during normal working hours.

With our best for a peaceful festive period and New Year, and we look forward to working with you in 2021.

Dear Ms Müller

The Editors assigned to your paper RSOS-201843 "Leaky barriers: leaky enough for fish to pass?" have now received comments from reviewers and would like you to revise the paper in accordance with the reviewer comments and any comments from the Editors. Please note this decision does not guarantee eventual acceptance.

Please submit your revised manuscript and required files (see below) no later than 21 days from today's (ie 23-Dec-2020) date. Note: the ScholarOne system will 'lock' if submission of the revision is attempted 21 or more days after the deadline. If you do not think you will be able to meet this deadline please contact the editorial office immediately.

on behalf of Dr Mark Smith (Associate Editor) and Pete Smith (Subject Editor)
openscience@royalsociety.org

Associate Editor Comments to Author (Dr Mark Smith):

Associate Editor: 1

Comments to the Author:

Both reviewers agree in their suggestions of major reviews and present several helpful suggestions. Scaling issues need to be addressed and clarified as should the relationship between the experiments conducted and the passage of fish through leaky barriers in the real world. This may require a re-focusing of the paper and some additional experimental details, which constitute a major revision.

Reviewer comments to Author:

Reviewer: 1

Comments to the Author(s)

The experiments are interesting and it is challenging to perform hydraulic experiments with living organisms. I do think there is a publishable study in here but, in its current form I have

concerns detailed below. The main one is that, to me at least, these experiments are highly abstract and not comparable to conditions in 'field scale' rivers – the scaling in these experiments does not appear to be meaningful for the fish, leaky dam or the hydraulics. Therefore, I have concerns about the relevance of the findings to leaky dams and fish in rivers. However, I think this is made into a bigger issue than it needs to be due to the pitching of the whole paper around leaky barriers. I would personally rework to focus more around fish behaviour in experimental conditions and how fish navigate around and through structures, and then have a much more brief mention at the end of the discussion that these results, with further work, may be of relevance for x,y,z reasons, one of which may very well be leaky dam design. I hope the comments below help the authors.

Page 1, Line 10 beginning "If global warming..." – should the reference be at the end of the sentence? Presumably the 220% is a prediction by the paper referenced earlier in this sentence.

Page 2, Line 25: there is a typo in this sentence.

Page 2, Line 38 beginning "The direction..." reads strangely to me.

Page 2, Line 46 - 50: I found this section a bit confusing. I understand it is a scale model but scaled to what? – I am guessing this isn't a Reynolds or Froude scaling approach but, some information on how the flow is scaled relative to barriers is needed if results are going to be used to inform field-scale work. In addition, are these flows scaled to a particular river? The discharge seems low relative to the depth, meaning the velocity is high (presumably around 0.3 m/s) – I cannot see how these conditions could be 'scaled up' to be relevant to field-scale river conditions.

I also wanted to see a breakdown of control flow conditions at this point but I noticed it came later. Might be worth referring to that table here too.

Page 2, Line 54: Given the condition is 80% bankfull it might be best to describe it as a "higher return period" than as a "higher flood return periods"

Page 3 – Figure 1 is useful but I still have difficulty visualising the barriers. A photograph of the barriers would be a useful addition.

Page 3, Line 30 - 44: "scaled from those installed in the field". As above, more information about this scaling is needed – what measurements of the field barriers were made and how was the scaling done? For example, does the 12.5 mm gap scale to any field measure? The manuscript states that two leaky barrier porosities were analysed but no information is given on where these come from and whether they are based on field measurements. I suspect these are completely abstract structures which is not a problem in itself but, I do think a refocusing of the manuscript away from 'recreating leaky barriers in rivers' is probably needed.

Page 5, line 51: Maybe I have misunderstood but the water depth is 15 cm yet the fish are approximately 12 cm long. Again, the scaling here seems way off to me – especially if this is meant to be bankfull flow. Given it is known the ability of salmon to overcome and navigate barriers is dependent on the ration of upstream and downstream water depth this strikes me as a problem. Plus, the 'gap' under the barrier was 5 cm high which again, doesn't strike me as a relevant fish-structure scaling for a real world situation. As above, this is much more of a problem if the purpose is to recreate river conditions, which appears to be the aim. It is much less of a problem if the focus is on fish behaviour more generally around porous and non-porous structures.

Page 7, line 54: I think it would be useful to define the 3 regions, particularly “under flow” – presumably this means the flow beneath the structure in the opening but confirmation of this would be good.

Page 7, line 55 beginning “In all cases...” this sentence is worded oddly to me.

Page 8, line 53: typo and strange characters in this sentence.

Page 10 – Figure 5 – I got really confused here but, I think the colours are mislabelled in the caption. Downstream is dark blue not green, etc. Also, should figure 5c be colour coded?

Page 11 line 38 – 45: I agree with this summary of the findings, which are interesting in informing about how fish interact with an abstract obstacle but I just do not see given the scaling of fish to barrier, and the depth to velocity scaling of the flow, how this can really be related to fish behaviour around field-scale, leaky barriers.

Page 12, line 27: What colour was the polythene wrapping?

Page 12, line 35: I don't find the colour argument that convincing. Through rheotaxis the fish will know which way is upstream. I do agree that the colour may influence their behaviour.

Page 12, line 40: There is a transition here from discussion about colour to overhanging logs. I would separate into 2 separate paragraphs

Page 12, line 46-57: I agree but none of this content is really informed by the experimental findings.

Page 13, line 8 – 12: I agree with all this and I think this is an interesting finding. The second half when referring to leaky barriers I find more problematic because there are so many confounding influences and because the results here are far from conclusive.

Reviewer: 2

Comments to the Author(s)

This paper presents an experimental study of flow fields and fish behaviour around leaky wooden dams of two different porosities and at two different discharges. Leaky wooden dams are increasingly being used in small headwater streams to contribute to "slowing the flow". I have concerns over a number of aspects of this study:

1. scaling of channel and wood dimensions (scaling is not adequately presented; are you Froude scaling or merely applying geometric similitude? We need more information about the field prototype to know)
2. scaling of fish relative to the above (how does the width and height of your model salmon compare to prototype salmon? How does this scale compare to your channel scale (width, depth)? How do either of these scales compare against your wood structure scale (both the elements and their spacing)? How do all these compare against the scale of the burst and/or sustained swim speed of your model salmon to prototype salmon?)
3. fish behaviour- generally speaking, a juvenile salmon doesn't really want to go up or downstream and it would be perfectly fine/happy if it never passed your barrier. If it was an anadromous salmonid (the next stage in the life cycle), it would swim seaward when the time was right, but given their size I don't think that would happen for another year. So the question is the extent to which "motivation" was dominated by the fright response of the fish to being prodded and whether fish were hiding underneath the porous barrier and exhibiting avoidance

behaviour when they swam upstream? Unfortunately, we cannot tell. I think that all prodded fish must be removed from your analyses in order to eliminate these possibilities.

4. the real-world applicability of leaky wooden dams preventing access to headwaters by adult salmon; leaky wooden dams are generally installed in headwater streams that are not reachable by adult salmon, so it is unclear how applicable your results are in the real world.

In addition to the above, i think that this study would benefit from inclusion of additional velocity data and exploration of the jet recovery downstream of structures. I have seen this data presented in non peer-reviewed outlets by your team, so it should not be too challenging to include it here; i think that the contrast between hydraulics of porous/non-porous barriers is of significant interest.

I have made some other edits and suggestions on the manuscript for your consideration (attached).

===PREPARING YOUR MANUSCRIPT===

Your revised paper should include the changes requested by the referees and Editors of your manuscript. You should provide two versions of this manuscript and both versions must be provided in an editable format: one version identifying all the changes that have been made (for instance, in coloured highlight, in bold text, or tracked changes); a 'clean' version of the new manuscript that incorporates the changes made, but does not highlight them. This version will be used for typesetting if your manuscript is accepted.

===PREPARING YOUR REVISION IN SCHOLARONE===

Author's Response to Decision Letter for (RSOS-201843.R0)

See Appendix B.

Decision letter (RSOS-201843.R1)

Dear Ms Müller,

It is a pleasure to accept your manuscript entitled "Leaky barriers: leaky enough for fish to pass?" in its current form for publication in Royal Society Open Science.

You can expect to receive a proof of your article in the near future. Please contact the editorial office (openscience@royalsociety.org) and the production office (openscience_proofs@royalsociety.org) to let us know if you are likely to be away from e-mail contact – if you are going to be away, please nominate a co-author (if available) to manage the proofing process, and ensure they are copied into your email to the journal.

on behalf of Dr Mark Smith (Associate Editor) and Pete Smith (Subject Editor)
openscience@royalsociety.org

Associate Editor Comments to Author (Dr Mark Smith):

Associate Editor

Comments to the Author:

All reviewer comments have been considered thoroughly. Extensive revisions have been made, especially to the framing of the paper and also in adding details on the geometric scaling, as requested. The authors have documented these changes carefully and clearly.

I am happy to accept this as it is (noting a few minor typographic errors have made their way into the edited proof that will no doubt be captured down the line).

Many thanks for undertaking such a thorough and detailed revision.

Follow Royal Society Publishing on Twitter: @RSocPublishing
Follow Royal Society Publishing on Facebook:
<https://www.facebook.com/RoyalSocietyPublishing.FanPage/>

Read Royal Society Publishing's blog:
<https://royalsociety.org/blog/blogsearchpage/?category=Publishing>

Appendix A**ROYAL SOCIETY
OPEN SCIENCE****Leaky barriers: leaky enough for fish to pass?**

Journal:	Royal Society Open Science
Manuscript ID	RSOS-201843
Article Type:	Research
Date Submitted by the Author:	14-Oct-2020
Complete List of Authors:	Müller, Stephanie; Cardiff University Cardiff School of Engineering, School of Engineering Wilson, Catherine; Cardiff University Cardiff School of Engineering, School of Engineering Ouro, Pablo; Cardiff University Cardiff School of Engineering, School of Engineering; The University of Manchester, School of Mechanical, Aerospace and Civil Engineering Cable, Joanne; Cardiff University Cardiff School of Biosciences, School of Biosciences
Subject:	environmental science < BIOLOGY, Environmental engineering < ENGINEERING AND TECHNOLOGY, ecology < BIOLOGY
Keywords:	Natural flood management, woody debris dam, acoustic Doppler velocimetry (ADV), fish behaviour, flooding
Subject Category:	Earth and Environmental Science

Author-supplied statements

Relevant information will appear here if provided.

Ethics

Does your article include research that required ethical approval or permits?:

Yes

Statement (if applicable):

All fish behavioural experiments were approved by Cardiff University Animal Ethics Committee and conducted under Home Office License PPL 303424 following the ARRIVE guidelines.

Data

It is a condition of publication that data, code and materials supporting your paper are made publicly available. Does your paper present new data?:

Yes

Statement (if applicable):

Data is available in the supplementary material.

Conflict of interest

I/We declare we have no competing interests

Statement (if applicable):

CUST_STATE_CONFLICT :No data available.

Authors' contributions

This paper has multiple authors and our individual contributions were as below

Statement (if applicable):

Concept and study design by CAMEW, JC and SM. Experiments were conducted, and manuscript drafted by SM. Data analyses and interpretation by SM, PO and CAMEW. All authors edited and revised the manuscript critically and approved the submitted version.

rsos.royalsocietypublishing.org

Research

Article submitted to journal

Keywords:

Natural flood management; woody debris dam; leaky barrier; acoustic Doppler velocimetry (ADV); fish behaviour; flooding

Author for correspondence:

Stephanie Müller

e-mail: MullerS1@Cardiff.ac.uk

Leaky barriers: leaky enough for fish to pass?

Stephanie Müller¹, Catherine A. M. E.

Wilson¹, Pablo Ouro^{1,3} and Joanne Cable²

¹Hydro-Environmental Research Center, School of Engineering, Cardiff University, Cardiff, CF24 3AA, UK

²School of Biosciences, Cardiff University, Cardiff, CF10 3AX, UK

³School of Mechanical, Aerospace and Civil Engineering, University of Manchester, Manchester, M13 9PL, UK

Leaky barriers, widely used in natural flood management, provide a sustainable and cost-effective supplement to traditional hard engineering methods for flood risk management. Sustainable, local materials are used to construct wooden leaky barriers across river channels in upper catchments aiming to slow down the flow, reduce flood peaks and attenuate the flow reaching downstream communities. These structures, however, present a potential fish migration barrier, yet little is known about their impact on hydrodynamics and fish passage. Here, we examined two scaled leaky barrier designs under 100% and 80% bankfull flow conditions in an open channel flume. These leaky barriers included a porous and a non-porous design, with the latter emulating the natural accumulation of brush, sediment and leaf material between log members over time. Flow visualization and velocity measurements recorded with acoustic Doppler velocimetry (ADV) characterized the flow field up- and downstream of the leaky barriers. Our fish behavioural studies revealed that juvenile salmon (*Salmo salar*) movement between downstream and upstream sections of the flume was inhibited by barrier design rather than discharge, influencing upstream fish passage and their spatial preference, indicating the importance of leaky barrier design criteria to facilitate fish movement.

© 2014 The Authors. Published by the Royal Society under the terms of the Creative Commons Attribution License <http://creativecommons.org/licenses/by/4.0/>, which permits unrestricted use, provided the original author and source are credited.

1. Introduction

Flooding causes significant socio-economic effects [1], likely to increase with more frequent, higher intensity rainfall events due to climate change [2,3]. If global warming increases mean temperatures by 4°C by 2080 as predicted by [4], this will increase global flood risk by 220%. There is therefore an urgent need to find sustainable solutions to attenuate the impact of flooding, shifting from traditional flood defences to an integrated flood risk management approach [5]. Such natural flood management (NFM) mitigation schemes [6–9] include working with natural processes and materials to slow down and store flood water [8].

Natural flood management measures can be subdivided into four categories: woodland, runoff, coastal and estuary, and river and floodplain management [6]. The latter involves river and floodplain restoration, offline storage areas, and the use of wood from surrounding areas to form leaky barriers [6] (otherwise known as logjams [10]). Over the last two decades, the use of woody debris and channel-spanning natural and particularly engineered leaky barriers have gained in popularity worldwide [11,12], but particularly in the UK [6,9,13,14] where a wide range of leaky barrier structures have already been installed as part of 60 DEFRA (Department of Environment, Food and Rural Affairs) funded NFM projects ([6,15]; e.g. Pickering, North East England; Holnicote, South West England; Shropshire, West England; Stroud, South West England; Great Triley Wood, South East Wales and Peeblesshire, South Scotland) [6,7,14]; although little is known about the associated hydrodynamics of these barriers. Leaky barriers, artificially introduced into upper catchments, span the river channel cross-section and are formed from wooden logs, fallen trees and branches. Due to their porous nature, these barriers allow flow through the structure and, in order to facilitate unimpeded baseflow and fish movement, they are also designed with a vertical gap between the barrier and the river bed. Under high flow conditions, the additional blockage created by these barriers causes the river to spill out onto the surrounding floodplains, making use of the floodplain storage which enhances filtration into the ground [16]. This, combined with the hydraulic roughness provided by floodplain vegetation and forest [17], slows down the movement of the water throughout the catchment and attenuates flow downstream [18]. The flood performance of multiple leaky barriers along a reach has been monitored at a range of field sites in the UK and results so far appear to be site-specific [6]. Along with a combination of other natural flood management measures, such barriers have been shown to reduce flood peak (Holnicote, South West England; Pickering, North East England; Stroud, South West England; Hampshire, South England), decrease and slow down downstream discharge (Belford, North East England; Guisborough, North East England; Great Triley Wood, South East Wales) and to increase upstream water level (Great Triley Wood, South East Wales) [14,19].

Besides their potential to mitigate the impact of flood events, leaky barriers can also improve water quality, divert flow and create new habitats [6]. Although leaky barriers are artificially formed by structures spanning the entire river width, their impact is often similar to natural accumulations of wooden logs and branches typically found along riverbanks in riparian woodland or in low velocity areas. By connecting the main channel with surrounding floodplains, wood accumulations can create seasonal wetlands [20], supporting lateral habitat connectivity [21], providing fish spawning and nursery grounds [22], as well as low velocity areas protecting fish from downstream displacement during high flows [23]. During flooding events, these structures are partially or completely submerged, altering the flow field and river morphology upstream and downstream by the creation of pools and other bed forms [24]; these low and high velocity zones enhance habitat complexity. Another important aspect is the creation of cover [18,25,26] as well as refugia for fish [18,25,27]. Small fish, in particular, can be attracted to coarse, branchy, complex wooden structures to seek shelter from predators, decreasing predator foraging success due to visual interference and prevention of entry [27]. Juvenile salmon particularly show increased survival when coarse, woody debris are present [27]. Nevertheless, fish responses to natural or the artificial introduction of wooden logs as in the case of leaky barriers may vary

depending on fish species and decomposition of the wood as well as other habitat characteristics [28].

So far, only a few studies have assessed the impact of leaky barriers used for flood mitigation on fish movement and habitat alteration. Based on existing knowledge about fish movement and habitat usage of target species found in Scottish rivers (salmon, trout, European eels and three-spined sticklebacks), guidelines have been developed for the use of such barriers in small streams in order to ensure fish migration [13]. These guidelines defined leaky barrier design requirements, namely minimum distance between river bed and lowest barrier edge, gap sizes within the barrier and maximum vertical barrier height. Such guidelines provide the first step in facilitating fish movement but they have not considered the interaction of leaky barriers with the aquatic ecosystem, including alterations in flow regime and hydraulics such as the interplay between floodplains and main channel usage, habitat alterations affecting shelter and food provision, or changes in predator-prey relationships. Here, we investigate the effect of two physical leaky barrier designs varying in structure porosity, comprising of a porous and non-porous barrier, facilitated with a vertical gap underneath. The non-porous physical design emulates the natural accumulation of branches, sediment and leaf material between the log members over time as the barrier becomes more water tight and therefore allow less flow through with time. Both barriers were scaled from those found in Wild Brook, Coverdale, Shropshire, UK, installed as part of the NFM project "Shropshire Slow the flow at Severn Tributaries" project to mimic their natural characteristics as close as possible, including dowel diameter, inter-dowel gaps and gap between bed and barrier, and are investigated under two flow conditions, representing bankfull and near bankfull flow, on channel hydrodynamics and fish behaviour through flume laboratory experiments.

2. Materials and Methods

(a) Flume set-up

Hydrodynamic and fish behaviour experiments were performed for two leaky barrier structure configurations in a recirculating open channel flume in the Hydro-Environmental Research Centre at Cardiff University, UK. The flume, presented in Figure 1 (a), is 10 m long (l_{flume}), 1.2 m wide (b_{flume}) and 0.3 m deep (h_{flume}). The direction of mean longitudinal flow was defined as positive in x direction defines as longitudinal coordinate, and y and z defined as lateral and vertical coordinates respectively. A longitudinal bed slope of $1/1000$ was applied to the flume. The flume comprises of a straight compound channel with rectangular main channel cross-section of width $b_{mc} = 0.6\text{ m}$, bankfull depth $h_{mc} = 0.15\text{ m}$ and floodplains of width $b_{fp} = 0.3\text{ m}$ on either side of the channel.

A Sigmund Pulsometer pump (Sigmund Pulsometer Pump Ltd., type AM20A) controlled water discharge and water surface elevation was adjusted by the tailgate weir located at the downstream end of the flume. Discharge (Q) and flow depth (h_0) remained fixed throughout the experiments. Prior to the installation of the leaky barrier structures, uniform, subcritical flow conditions were established for bankfull (Q_{bf}) and 80% bankfull ($0.8Q_{bf}$) flow conditions relating to a discharge of $0.028\text{ m}^3/\text{s}$ and $0.022\text{ m}^3/\text{s}$ and a water depth of 0.15 m and 0.13 m, respectively. These conditions represent the control treatment. Bankfull flow condition refers to the maximum discharge capacity of the main channel and therefore the greatest flow rate contained within the main channel before flow inundates on to floodplains. Hence, 80% bankfull flow condition refers to 80% of maximum discharge capacity of main channel and was selected to represent higher flood return period than bankfull flood events. The installation of the leaky barrier structures resulted in a change in water surface profile generating gradually varied flow conditions. Flow depth was measured using a point gauge.

Figure 1. Schematic representing the experimental set-up showing a straight, rectangular compound channel of width $b_{mc} = 0.6$ m, bankfull height $h_{mc} = 0.15$ m and floodplains (*fp*) on either site of the main channel (*mc*) of width $b_{fp} = 0.3$ m. The leaky barrier structures were placed within the main channel at approximately 5 m downstream of the flume inlet.

(b) Leaky barrier structures

The leaky barrier structures, **scaled** from those installed in the field, were constructed using individual wooden dowels of diameter $d = 25$ mm, fixed in the main channel using silicon adhesive and located approx. 5 m downstream of the flume inlet, as shown in Figure 1 (b). Each dowel row comprised eight dowels in longitudinal x direction, with a barrier length (L_s) of 0.2 m and height (H_s) of 0.1 m. A vertical gap (b_0) of 50 mm remained between the lowest dowel edge and flume bed, designed to mimic field designs with similar cross-sectional flow blockage, allowing passage of baseflow and fish movement. A vertical distance b of 12.5 mm was maintained between the dowels allowing flow through. Two leaky barrier structure porosities, were analysed by comparing a porous structure against a non-porous structure simulating the natural accumulation of sediment, leaf material and woody debris and therefore the clogging of the barrier. The latter structure was constructed by wrapping the external dowels in orange polythene to prevent flow through the structure whilst the 50 mm gap underneath the leaky barrier structure remained present. Due to the presence of the vertical gap underneath both barriers, the channel cross-section is still deemed as porous. Here, we only concentrate on the porosity of the barrier structure itself.

For each flow discharge condition, two leaky barrier structures were tested, namely non-porous and porous, and one control condition, i.e. no structure present. Physical and hydraulic characteristics for each leaky barrier are presented in Table 1, including cross-sectional blockage area A_s , cross-sectional blockage ratio A_0 , being the ratio between structural frontal area A_s and flow area $A = H_{mc}b_{mc}$ as well as solid volume fraction $\phi = V_{structure}/V_{control}$, where $V_{structure}$ is the volume occupied by the solid leaky barrier, defined as $\pi(d/2)^2 n_{dowels} b_{mc}$, and $V_{control} = H_{mc}L_s b_{mc}$ is the volume for control conditions. The structural porosity of the porous design is defined as $\Phi = V_{pore}/(V_{pore} + V_{solid})$, being V_{pore} the equivalent porosity volume resulting from the difference between $V_{control}$ and $V_{structure}$. Hydraulic characteristics are calculated for upstream (1) and downstream (2) locations, such as mean flow depth (H_1 and H_2), which was calculated by building the average of all water elevation measurements upstream and downstream, respectively. Additionally, the difference between up- and downstream mean flow depths ($\Delta H = H_2 - H_1$) and bulk velocity (U_{01} and U_{02}) were computed. Reynolds number was calculated as $Re = U_0 R_H / \nu$, with ν denoting the fluid kinematic viscosity and R_H denoting

the hydraulic radius. Froude number is $Fr = U_0/\sqrt{gH_s}$ with g denoting gravity acceleration for the leaky barrier cases whilst for the control case is $Fr = U_0/\sqrt{gR_H}$.

Table 1. Details of the leaky barriers structural characteristics and flow conditions upstream (1) and downstream (2) of the structure, including cross-sectional blockage area (A_s), cross-sectional blockage ratio (A_0), solid volume fraction (ϕ) and structural porosity (Φ), mean flow depth (H_1 and H_2), difference between up- and downstream mean flow depth (ΔH , with * indicating overbank flow in column H_1), bulk velocity (U_{01} and U_{02}), Reynolds number (Re_1 and Re_2) and Froude number (Fr_1 and Fr_2) based on hydraulic radius.

	Treatment	A_s [m ²]	A_0 [%]	ϕ [%]	Φ [%]	H_1 [mm]	H_2 [mm]	ΔH [mm]	U_{01} [m s ⁻¹]	U_{02} [m s ⁻¹]	Re_1 [-]	Re_2 [-]	Fr_1 [-]	Fr_2 [-]
100% Q_{bf}	Control	-	-	-	-	145.9	-	-	0.32	-	31,367	-	0.38	-
	Non-Porous	0.06	66.67	66.67	-	172.3*	139.6	32.7	0.24	0.33	18,127	31,845	0.25	0.29
	Porous	0.05	50.00	39.29	41.1	158.4*	142.4	16.0	0.28	0.33	18,458	31,647	0.31	0.28
80% Q_{bf}	Control	-	-	-	-	132.3	-	-	0.32	-	25,424	-	0.24	-
	Non-Porous	0.06	66.67	66.67	-	157.5*	124.2	33.3	0.22	0.30	14,521	25,931	0.25	0.27
	Porous	0.05	50.00	39.29	41.1	140.5	131.0	9.5	0.26	0.28	13,852	25,522	0.22	0.25

(c) Velocity measurements and flow visualization

Flow velocities were measured upstream and downstream of the leaky barriers using a sideways-looking acoustic Doppler velocimeter (ADV) (Nortek Vectrino) at a sampling rate of 200 Hz. To ensure sufficient data quality and to capture a representative sample of high-frequency turbulent fluctuations, measurements were conducted over 5-20 min. The water was seeded using Spherical[®] 110P8 hollow glass spheres (Potters Industries LLC) with a mean particle size of 11.7 μ m and a specific gravity of 1.10 g/cc. During measurements, a signal-to-noise ratio (SNR) of at least > 15 dB and a correlation $> 70\%$ were maintained by adding seeding material to enhance signal quality. Velocity profiles were taken at two locations upstream ($x/b_0 = -10.4$ and -1.4) and two downstream ($x/b_0 = 0.8$ and 10.6) for the two barriers and for both flow conditions, as indicated in Figure 2. For the control treatment, velocity measurements were conducted over a single vertical profile at $x/b_0 = 0.8$. The velocity profiles consisting of 20-26 point measurements with a vertical resolution of 5 mm. Velocity data were filtered and post-processed using Matlab (R2018b and R2019b). In a first pre-filtering step, data with insufficient SNR and correlation were removed before the data were despiked using an open-source toolbox [29,30]. Time-averaged velocities are calculated from the time history at every measured point and are denoted by an overbar. Additionally, flow patterns of the near wake were visualized for both leaky barrier structures and flow conditions using Fluorescent Fwt red (Cole-Parmer Instrument Company Ltd), which was injected for flow visualisation at the centreline upstream of the structures at a range of elevations. A GoPro Hero 5 underwater camera positioned at the left-hand side of the main channel to record the dispersion of the dye tracer.

(d) Fish passage experiment

Fish passage behaviour tests were conducted between January 21st and February 1st 2019 between 7.30am and 10pm. Atlantic salmon (*Salmo salar*, mean \pm s.d. mass 12.3 g \pm 5.4 g, mean \pm s.d. standard length 93.3 mm \pm 13.6 mm), sourced from Kielder Salmon Centre were maintained within the Cardiff University Aquarium at $14 \pm 1^\circ$ C. Atlantic salmon are anadromous, they hatch in rivers and streams, migrate to the sea to mature, and then salmon migrate back to their birthplace to spawn. Growing up in fast flowing, clean and well oxygenated freshwater, these fish demonstrate positive rheotaxis, orientating themselves against the flow direction (Kalleberg (1958) in [31]). Swimming tests with juveniles (29.9 \pm 0.9 g, standard length 123 \pm 16 mm)

Figure 2. Schematic representing the position of the ADV measurements and fish behavioural trials test section. In total, four velocity profiles (upstream: $x/b_0 = -10.4$ and -1.4 ; downstream: $x/b_0 = 0.8$ and 10.6) were measured starting approximately 10 mm above flume bed until 30 mm beneath the water surface with a vertical resolution of 5 mm.

determined a critical swimming speed at 0.93 ± 0.103 m/s, but this swimming performance is strongly dependent upon fish size [32].

The experimental test section, shown in Figure 2, was 1.6 m long and bounded by a plastic garden mesh with square holes of size 5 mm \times 5 mm, spanning the entire width of the flume, at approximately 0.7 m ($14b_0$) up- and downstream of the structure. For convenience in the analysis, the test section was divided into three spatial zones: upstream ($\Delta x = 0.7$ m), downstream ($\Delta x = 0.7$ m) and structure ($\Delta x = 0.2$ m). A GoPro Hero 5 underwater camera was positioned $x = 0.7$ m upstream of the barrier, on the main channel centreline outside the mesh restriction and pointed in the downstream direction. The water was dechlorinated using Seachem Prime Concentrated Conditioner and chilled to 14°C.

Each fish was gently transferred to the flume and given a 15 min acclimatisation period including a 2 min incremental increase in discharge over the first 10 min up to the test discharge level, followed by a 5 min acclimatisation at this test discharge (22 l/s or 28 l/s) before each trial commenced ($n = 16$ for porous, 80% bankfull discharge; otherwise $n = 14$ per treatment). Each trial lasted 10 min where individual fish were released at the most downstream end of the test section along the centreline of the main channel. Treatment order could not be randomized because of the construction method of the installed barriers. The porous structure was tested first, followed by non-porous structure and control condition. For each treatment, 80% bankfull discharge was tested prior to 100% bankfull flow condition. Human intervention took place only in cases where fish remained stationary in the furthest downstream transect and refused to swim. In this case, fish were gently nudged with the handle of the net, and in the 3 cases this occurred this stimulus worked for 26 of the fish. The nine non-responding fish were excluded from the analysis (included in analyses: $n = 10$ for control 80% Q_{bf} , $n = 12$ for control 100% Q_{bf} , $n = 13$ for porous 80% Q_{bf} , otherwise $n = 14$). During each test, time spent in each zone (upstream, downstream or underneath barrier) and number of upstream passes was recorded manually using stopwatches, and the GoPro Hero 5 underwater camera that was positioned upstream of the test section at the centre of the main channel, facing in downstream direction (Figure 2) as compound channel walls and changes in flow depth did not allow side or top view.

Statistical analysis was conducted using R v.3.6.3 statistical software. Spatial preference was analysed using a separate general linear model (GLM) with gaussian distribution and identity link for each spatial zone, allowing the investigation of the difference in mean between time spent upstream, downstream and underneath the structure (time proportion as dependent variable) and leaky barrier treatment as well as flow condition (independent variables). Association between number of upstream passes per fish (dependent variable) and leaky barrier as well as flow condition (independent variable) was tested using a GLM with Poisson distribution and identity as link function. A binomial GLM with link function logit was conducted to analyse

potential associations between flow condition as well as barrier and upstream passed fish and flood plain usage which ~~have been~~ reported as categorical variables (passed/not passed or used/not used). Non-significant variables were stepwise removed from the statistical analysis and residuals were used to assess the suitability of the tests. P-value significance was taken at 0.05.

3. Results

Channel hydrodynamics and fish behaviour were quantified for two leaky barrier structures and a control situation (no barrier) and two flow conditions to determine the impact of flow conditions and structure on spatial preference and upstream passage of juvenile Atlantic salmon.

(a) Hydrodynamics

Normalised time-averaged longitudinal velocity (\bar{u}/U_{01}) results for the main channel hydrodynamics measured for the control situation at $x/b_0 = 0.8$ (a), and for the non-porous barrier (b) and porous barrier (c) profiles for 100% (green) and 80% bankfull (blue) flow conditions at the four x/b_0 locations indicated in Figure 2 are presented in Figure 3. The measured velocity profiles for the control scenario represent typical open-channel flow conditions following a logarithmic distribution, with slightly higher normalised mean longitudinal velocities found for 80% bankfull discharge, indicating higher distribution of momentum in the centre of the channel (Figure 3 (a)). This is caused by ~~the~~ sidewall effects and the associated hydrodynamics re-distributing ~~the~~ momentum towards the channel centre under the 80% bankfull discharge.

Upstream velocity profiles for the non-porous and porous barriers are shown on the left-hand side of Figure 3 (b) and (c), respectively. Furthest upstream ($x/b_0 = -10.4$) of the barriers, velocity profiles still follow a near-logarithmic distribution similar to that recorded under control conditions. Immediately upstream of the structure ($x/b_0 = -1.4$) higher values of \bar{u}/U_{01} occurred for the 80% bankfull flow conditions regardless of the leaky barrier structure. For the non-porous design, however, there is higher momentum flow going through the bottom gap, changing the velocity distribution with the highest values occurring at mid-gap height ($0.5b_0$) and progressively decreasing towards the water surface. Such changes in the longitudinal velocity distribution for 80% bankfull flow conditions are more subtle for the porous barrier as a result of a decreased flow blockage as this structure allows ~~flow through~~. For the non-porous case, due to its blocking-nature (Figure 3 (b)) there is a 20% increase in upstream flow depth for both flow conditions compared to control conditions, see Table 1. This resulted in overbank ~~and overtopping~~ flows ~~occurring~~ for both discharges, which led to lower in-channel mean longitudinal velocities upstream of the barriers. For the non-porous structure, the main-channel flow depth exceeded bankfull flow depth by 15% and 5% for bankfull and 80% bankfull discharges, respectively, whilst for the porous ~~leaky~~ structure this only increased by 8% and 6% for bankfull and 80% bankfull discharge, respectively, with overbank flow only observed for bankfull flow conditions (see Table 1).

Downstream velocity profiles are shown on the right-hand side of Figure 3 (b) and (c) for the non-porous and porous barriers, respectively. Immediately downstream of the non-porous leaky barrier ($x/b_0 = 0.8$), the maximum \bar{u}/U_{01} is found at approximately one third of the gap height ($0.33b_0$) and increased 2.7 and 2.0 times compared to values at $x/b_0 = -1.4$ (Figure 3 (b)) for 100% and 80% bankfull flow, respectively. The maximum \bar{u}/U_{01} was 10% higher for the 80% bankfull discharge than for bankfull conditions, as in the latter case, the upstream flow spills onto the floodplains and overtops the barrier, ~~therefore~~ momentum ~~is re-distributed and taken away~~ from the main channel ~~region~~ and more specifically, from the "under flow" region. In all cases, profiles of \bar{u}/U_{01} show a progressive decrease with increasing ~~the~~ elevation in the water column. Immediately downstream of the porous leaky barrier, the maximum longitudinal velocity only increased by approximately 1.8 times compared to $x/b_0 = -1.4$ for both flow condition, ~~respectively~~ (Figure 3 (c)). Longitudinal mean velocities were still slightly higher for

Figure 3. Mean longitudinal velocity profiles obtained under 80% (blue) and 100% (green) bankfull flow condition for control (a), non-porous (b) and porous (c) leaky barriers.

80% bankfull flow, likely due to the overbank flow observed under bankfull conditions, as well as the increased flow through the barrier. A notable feature in the wake of the leaky barrier is a second peak featuring a slight increase in longitudinal mean velocity at the lowest inter-dowel gap, i.e. $z/b_0 = 1.5$, as a result of flow going through the porous leaky barrier. With increasing downstream distance, longitudinal velocities start to recover. Far downstream, at $x/b_0 = 10.6$, the difference in \bar{u}/U_{01} between discharges was more pronounced for the non-porous barrier with higher longitudinal mean velocities under the bankfull flow. In contrast, velocity recovery was found to be independent of discharge for the porous leaky barrier, likely due to the reduced overtopping flow.

As the pointwise measured velocity profiles only provide information about the time-averaged velocity statistics, flow visualization was also used to observe the instantaneous flow field in the wake of the leaky barriers. Dependent on the barrier, the downstream flow field was characterised by high momentum flow, through flow and over topping flow which can be seen in Figure 4. The significant difference in overtopping flow observed for both barriers is shown in Figure 4 (a) and (b) for 100% bankfull flow. In particular, in the case of the non-porous barrier, the overtopping flow was observed to plunge along the barrier's edge, joining the high momentum flow from under flow region (see Figure 4 (c)). Less strong overtopping flow was present for the porous leaky barrier due to the flow through the barrier (see Figure 4 (d)). No significant overtopping flow was observed at 80% bankfull flow conditions for both leaky barriers.

Figure 4. The near wake of both leaky barriers (shown on the left-hand side of each image) was visualised using fluorescent dye, injected at different heights upstream of the barrier. Flow alterations observed include overtopping flow (non-porous (a) and porous (b)), high momentum flow (c) and flow through the porous leaky barrier (d). The fluorescent dye is indicated in red with flow direction being left to right.

(b) Fish behaviour

Fish behaviour results were analysed in terms of time fish spent downstream, upstream and underneath the barrier as well as percentage of fish passing from the downstream into the upstream region, and mean number of upstream passes per fish (Figure 5). While no significant changes between 80% and 100% bankfull flow were observed for spatial preference and upstream fish passes, the barrier design did impact fish behaviour.

Mean proportion of time fish spent downstream, upstream and underneath the barrier after being released are shown in Figure 5 (a) for 80% bankfull (left) and 100% bankfull (right) flow conditions. Spatial preference was significantly more impacted by barrier porosity (GLM, all $p < 0.001$) than by an increase in discharge (GLM, all $p > 0.001$). Independent of the flow condition, fish spent more time downstream when no leaky barrier (control) was placed inside the test section (71 and 77% for 80% bankfull and bankfull discharge, respectively). Time spent downstream significantly differed between control condition and barriers present (GLM, porous: $p = 0.0097$ and non-porous: $p = 0.0057$), but also amongst barriers (GLM, $p = 0.0858$). In contrast, time spent upstream only significantly differed from the control case when the non-porous barrier was present (GLM, $p = 0.0239$). In case of the non-porous leaky barrier, fish spent similar time upstream (48% and 41%) and downstream (50% and 44%) for 80% and 100% bankfull flow condition, respectively. An increase in time spent in the upstream section of 100% and 78% compared to control condition was observed for 80% and 100% bankfull flow, respectively. Time

spent underneath the barrier increased from 2% to 15% when increasing the discharge but did not differ significantly from the control condition (GLM, $p=0.341$). Conversely, in the presence of the porous leaky barrier, fish spent most time underneath the barrier, demonstrated by 69% and 70% for 80% bankfull and 100% bankfull discharge, respectively, which significantly differed from the control condition (GLM, $p < 0.001$) as well as from what was observed for the non-porous barrier (GLM, $p < 0.001$). Similar to what was observed for the non-porous barrier, fish spent equal time upstream (15%) and downstream (16%) under 80% bankfull flow condition, however, under 100% bankfull flow time spent downstream increased to 29% while time spent upstream decreased to 1%.

Figure 5. Summary of fish behavioural test showing (a) average time fish spent downstream (green), beneath the structure (light blue) and upstream (blue) under control, non-porous and porous barrier for 100% (left) and 80% (right) bankfull flow conditions. Percentage of fish passing from downstream area into upstream area is presented in (b) for 80% (blue) and 100% (green) bankfull flow conditions. Mean upstream passes per fish are shown in (c) with error bars representing standard deviation.

The percentage of fish passing at least once from the downstream into the upstream region for 80% (blue) and 100% bankfull (green) flow conditions is presented in Figure 5 (b). No significant association was found between flow condition and percentage of upstream passing fish (GLM, $p=0.9667$), however, a significant difference between control situation and the presence of the barriers was noted (GLM, $p=0.0026$). While in the absence of a leaky barrier (control), 60% and 50% of the tested fish passed at least once into the upstream region under 80% and 100% bankfull flow condition, respectively, a higher percentage of fish passed upstream when a non-porous leaky barrier was present (GLM, $p=0.0665$) and smaller percentage of fish passed upstream under the presence of the porous leaky barrier (GLM, $p=0.1573$). The percentage of fish passing upstream was significantly different between both leaky barriers (GLM, $p=0.0012$).

As every fish was able to pass multiple times from the downstream region into the upstream region, Figure 5 (c) presents the mean number of passes per fish. Despite no significant association between number of upstream passes per fish and flow condition (GLM, $p=0.9963$), the mean number of upstream passes per fish significantly differed when a barrier was present (GLM, $p=0.0174$). Under the control treatment, every fish passed on average 1.4 and 0.67 times from downstream to upstream under 80% and 100% bankfull discharge, respectively. This number decreased not significantly in the presence of a non-porous leaky barrier (0.71 and 0.93 times for 80% and 100% bankfull discharge; GLM, $p=0.5138$), but even more so when a porous barrier was present which led to a significant difference to the control condition (0.31 and 0.43 time for 80% and 100% bankfull discharge; GLM, $p=0.0096$). In addition, a significant difference in mean passes per fish was found amongst both barriers (GLM, $p=0.0297$). Highest variation in mean passes per fish was found for the control treatment.

When increasing the discharge to bankfull flow or under blockage (i.e. non-porous structure) conditions, upstream water level rose, inundating both floodplains on either sides of the main channel and therefore, opening potential, new habitat. Interestingly, under these conditions, a small, not significant, minority of fish (7% for non-porous barrier, both discharges and 17% for control, 100% bankfull discharge) used the additional space by swimming onto the floodplains. Hence, no significant association was found between floodplain usage and discharge (GLM, $p=0.3304$) and leaky barriers presence (GLM, $p=0.1621$).

4. Discussion

We have shown that two leaky barrier designs did not prevent fish movement but did impact fish behaviour when compared against the no barrier, free flow case. Interestingly, the difference in discharge was not the decisive component impacting fish behaviour, instead the physical design of the leaky barrier and the structure's porosity was more important. Spatial preference, percentage of upstream passing fish and passage number varied amongst the tested leaky barriers which may be linked to a range of reasons including hydrodynamic alterations of the surrounding flow field, visual cues or the provision of shelter.

Under high flow conditions, leaky barriers aim to reconnect the main channel flow zone with the adjacent floodplain zone. By doing so, water backs up and spills onto the floodplains and inundates them, creating new habitat for aquatic organisms but also supporting upstream nutrient and sediment exchange [21]. Floodplains often contain wood in the form of logs, trees, branches and brush with high densities of macro-invertebrates and therefore potentially provide additional food sources for fish [34] but also spawning and nursing grounds during high flow periods [21, 22]. However, not all flow conditions lead to the inundation of floodplains. It strongly depends on discharge and channel cross-section as well as physical properties of the leaky barrier. Although, in our experiment fish explored the floodplain regions characterised by low velocities, this may not be the case in the field with natural ground comprising mud and debris.

When comparing 100% and 80% bankfull flow conditions, no differences in fish behaviour were observed, likely due to the fact that velocity magnitudes did not vary greatly between the analysed flow conditions. In addition, salmon are relatively strong swimmers, well adapted to high flow velocities, typically found in upland streams. A higher increase in absolute longitudinal

mean velocities was observed for the lower discharge, less impacted by overbank flow, which mostly occurred at higher discharge and barrier blockage. These barriers not only impact upstream and downstream flow depth, they strongly alter the velocity distribution in its vicinity. A range of flow variations were detected in our experiments including overtopping flow, flow through the porous barrier, and high momentum flow coming from underneath the leaky barrier. These high longitudinal velocities beneath the barrier prevent this gap and therefore the channel cross-section from blocking by sediment, leaf material and woody debris accumulation when installed in the field, and did not prevent fish from passing into the upstream region, indeed fish actually sought shelter underneath. For the non-porous leaky barrier, in particular, a larger number of fish passed upstream and spent more time in this flow volume, despite the slightly higher longitudinal mean velocities. This may indicate that higher momentum flow provides a clearer cue for fish of where to pass. Such flow acceleration may not only impact fish behaviour but also alter the river bed by creating scour and promoting sediment transport, potentially leading to the exposure or smothering of eggs at spawning grounds as well as survival of benthic macro-invertebrates [18,35]. Woody debris have the ability to create complex habitats with riffles and pools, fostering an increase in fish abundance, species richness [18,36] and biomass (e.g. for largemouth bass [37]). Additional hydrodynamic measurements were conducted in [38]. Similar to the present study, the impact of first and second order flow statistics was smaller compared to the physical design of the leaky barrier.

Besides the flow alterations caused by the non-porous and porous leaky barriers, physical appearance may impact the spatial preference and passage of fish. In the current study, the main physical difference between the tested barriers was the polythene wrapping, which prevents through flow, but also created a more unnatural, coloured obstacle compared to the porous structure which clearly shows natural elements in form of wooden dowels. Depending on species, fish are able to differentiate colours and are attracted to different colours [39]. For instance, while bluegill sunfish and young carp react more towards red [39,40], Japanese marine fish species show greater preference for blues and greens [41]. Although several salmonids have been recorded to possess colour vision (e.g. *masusalmon*) [42], so far little is known about their attraction to colours. Thus, the coloured wrapping of the non-porous leaky barrier might have acted as a visual cue, guiding fish upstream. In this context, it should be noted that fish perception of the barrier colour may have been compromised due to variations in ambient light. The natural appearance of the porous leaky barrier may increase the attractiveness of this structure as a fish shelter, potentially being the reason why fish spent more time underneath the porous leaky barrier and therefore passed less often upstream. Overhanging logs and complex accumulations of wood and other debris are an important source of cover in rivers, which provides habitats for different species [18]. For instance, strong preference for overhead cover has been reported for Atlantic salmon at lower temperatures [43]. Besides cover, complex woody debris accumulation are an important refuge for small fish in case of predators by causing visual interference and entry prevention [27]. When comparing natural occurring complex woody debris structures against installed wooden structures, largemouth bass selected both structures at a similar rate [44].

In general, leaky barriers do not present a barrier to fish movement if certain design criteria are fulfilled, such as provision of a gap underneath the structure allowing unimpeded base flow and fish passage [13]. Importantly, regular maintenance of these barriers is required to prevent further blockage by driftwood, debris, sediment, leaf material or inorganic materials, leading to the creation of a physical, solid barrier to fish movement. Such field monitoring may also increase the lifespan of the barrier, but all amount to additional costs. Alternatively, the public could be engaged to document leaky barrier states by submitting photographs and/or being involved in community conservation projects to clear barrier of debris after a flood event. In the meantime, further research will be needed to assess blockage and structural decay over time. Generalisation of the current results, however, is limited as the study was only performed on one species of a particular size category, under strong lighting conditions in a simplified environment. In their natural environment, fish at different life stages may respond differently.

5. Conclusion

The impact of two leaky barrier designs on channel hydrodynamics and fish behaviour (*Salmo salar*) were experimentally investigated for two flow conditions. We show that leaky barrier porosity, rather than discharge, was the decisive component impacting upstream fish passage and spatial preference. Fish movement was influenced by porosity, with more fish undergoing upstream passage for the non-porous design compared to its porous counterpart. This expands our current state of knowledge on leaky barrier design in relation to fish and we recommend including porosity as a design parameter. This study, together with further research, will enhance design and delivery of leaky barriers as green, eco-friendly hydraulic structures used for natural flood management while ensuring the mitigation of flooding, maintaining habitat and enhancing connectivity for aquatic organisms.

Ethics. All fish behavioural experiments were approved by Cardiff University Animal Ethics Committee and conducted under Home Office License PPL 303424 following the ARRIVE guidelines [33].

Data Accessibility. Data is available on request.

Authors' Contributions. Concept and study design by CAMEW, JC and SM. Experiments were conducted, and manuscript drafted by SM. Data analyses and interpretation by SM, PO and CAMEW. All authors edited and revised the manuscript critically and approved the submitted version.

Competing Interests. The authors declare they have no competing interests.

Funding. This research was funded as part of the Water Informatics Science and Engineering Centre for Doctoral Training (WISE CDT) [EP/L016214/1] from the Engineering and Physical Science Research Council (EPSRC).

Acknowledgements. We thank Paul Leach, Steven Rankmore, Gareth Castle, Valentine Muhawenimana and Jelena Nefjodova for technical assistance and Rhi Hunt for providing statistical advice.

References

1. Santos S, Bender S, Schaller M. 2013 The European Floods Directive and Opportunities offered by Land Use Planning. Climate Service Center Germany An Institution of Helmholtz-Zentrum Geesthacht.
2. Jia G, Shevliakova E, Eduardo ANP, De Noblet-Ducoudre N, Richard H, House J, et al. 2019 International Panel on Climate Change, Special report: special report on climate change and land, Chapter 2, Land-Climate Interactions, Final Government Distribution.
3. Lehmann J, Coumou D, Frieler K. 2015 Increased record-breaking precipitation events under global warming. *Climate Change* 132(4):501–515; doi:10.1007/s10584-015-1434-y.
4. Alfieri L, Feyen L, Dottori F, Bianchi A. 2015 Ensemble flood risk assessment in Europe under high end climate scenarios. *Global Environmental Change* 35:199–212; doi:10.1016/j.gloenvcha.2015.09.004.
5. European Environment Agency. 2010 Mapping the impacts of natural hazards and technological accidents in Europe An overview of the last decade. Copenhagen: European Environment Agency; doi:10.2800/62638.
6. Burgess-Gamble L, Ngai R, Wilkinson M, Nisbet T, Pontee N, Harvey R, et al. 2018 Working with Natural Processes - Evidence Directory. Environment Agency - Flood and Coastal Risk Management Research.
7. Scottish Environment Protection Agency. 2015 Natural Flood Management Handbook.
8. Scottish Environment Protection Agency. 2013 Identifying Opportunities for Natural Flood Management.
9. Mott N. Managing Woody Debris in Rivers, Streams & Floodplains. 2006 Staffordshire Wildlife Trust, UK.
10. Wohl E. 2017 Bridging the gaps: An overview of wood across time and space in diverse rivers. *Geomorphology* 279:3–26; doi:10.1016/j.geomorph.2016.04.014.
11. Strosser P, Delacámara G, Hanus A, Williams H, Jaritt N. 2015 A guide to support the selection, design and implementation of natural water retention measures in Europe – Capturing the multiple benefits of nature-based solutions. European Commission; doi:10.2779/761211.
12. Reich M, Kershner JL, Wildman RC. 2003 Restoring streams with large wood: a synthesis. *American Fisheries Society Symposium* 37:355–366.
13. Dodd JA, Newton M, Adams CE. 2016 The effect of natural flood management in-stream wood placements on fish movement in Scotland. Scotland’s Centre of Expertise for Water (CREW).
14. Forest Research. 2007 The Robinwood Robinlood report: Evaluation of Large Woody Debris in Watercourses.
15. Arnott S, Burgess-Gamble L, Dunsford D, Webb L, Johnson D, Andison E., et al. 2018 Monitoring and evaluating the DEFRA funded Natural Flood Management projects. Environment Agency.
16. Kay AL, Old GH, Bell VA, Davies HN, Trill EJ. 2019 An assessment of the potential for natural flood management to offset climate change impacts. *Environmental Research Letters* 14(4); doi:10.1088/1748-9326/aafdb.
17. Piégay H. 1997 Interactions between floodplain forests and overbank flows: data from three piedmont rivers of southeastern France. *Global Ecology and Biogeography Letters* 6(3/4):187–196; doi:10.2307/2997732.
18. Dolloff CA, Warren Jr ML. 2003 Fish relationship with large wood in small streams. In: Gregory SV, Boyer KL, Gurnell AM, editors. The Ecology and Management of Wood in World Rivers. *American Fisheries Society Symposium* 37:179–193.
19. Rose S. 2015 From source to sea natural flood management | the Holnicote experience. Project code RMP 5508 - Multi-Objective Flood Management Demonstration Project - The National Trust Holnicote. National Trust.
20. Zalewski M, Magorzata M, Bayley PB. 2003 Fish relationships with wood in large rivers. In: The Ecology and Management of Wood in World Rivers. *American Fisheries Society Symposium* 37.

21. Burgess OT, Pine III WE, Walsh SJ. 2012 Importance of floodplain connectivity to fish populations in the Apalachicola River, Florida. *River Research and Applications* 29(6):718–733; doi:10.1002/rra.2567.
22. Sheaffer WA, Nickum JG. 1986 Backwater areas as nursery habitats for fishes in pool 13 of the Upper Mississippi River. *Hydrobiologia* 136:131–139; doi:10.1007/BF00051510.
23. Franssen NR, Gido KB, Guy CS, Tripe JA, Shrank SJ, Strakosh TR, et al. 2006 Effects of floods on fish assemblages in an intermittent prairie stream. *Freshwater Biology* 51:2072–2086; doi:10.1111/j.1365-2427.2006.01640.x.
24. Fausch KD, Northcote TG. 1992 Large Woody Debris and Salmonid Habitat in a Small Coastal British Columbia Stream. *Canadian Journal of Fisheries and Aquatic Sciences* 49:682–693; doi:10.1139/f92-077.
25. Senter AE, Pasternack GB. 2011 Large wood aids spawning Chinook Salmon (*Oncorhynchus Tshawytscha*) in marginal habitat on a regulated river in California. *River Research and Applications* 27:550–565; doi:10.1002/rra.1388.
26. Opperman J, Merenlender A, Lewis D. 2006 Maintaining wood in streams: a vital action for fish conservation. *eScholarship University of California* (8157); doi:10.3733/ucanr.8157.
27. Sass GG. 2009 Coarse woody debris in lakes and stream. In: Likens GE, editor. *Encyclopedia of inland waters* vol. 1. Oxford: Elsevier; p. 60–69; doi:10.1016/B978-012370626-3.00221-0.
28. Archer MW, Pracheil BM, Otto AE, Pegg MA. 2019 Fish community response to in-channel woody debris in a channelized river system. *Journal of Freshwater Ecology* 34(1):351–362; doi:10.1080/02705060.2019.1614103.
29. Mori N. 2020. Despiking. MATLAB Central File Exchange.
30. Mori N, Suzuki T, Kakuno S. 2007 Noise of acoustic Doppler velocimeter data in bubbly flow. *Journal of Engineering Mechanics* 133(1):122–125; doi:10.1061/ASCE0733-93992007133:1122.
31. Thorpe JE, Morgan RIG 1978 Periodicity in Atlantic salmon *Salmo salar* L. smolt migration. *Journal of Fish Biology* 12:541–548; doi:10.1111/j.1095-8649.1978.tb04200.x.
32. Palstra AP, Kals J, Böhm T, Bastiaansen JWM, Komen H. 2020 Swimming Performance and Oxygen Consumption as Non-lethal Indicators of Production Traits in Atlantic Salmon and Gilthead Seabream. *Frontiers in Physiology* 11:759; doi:10.3389/fphys.2020.00759.
33. Kilkenny C, Browne WJ, Cuthill IC, Emerson M, Altman DG. 2010 Improving Bioscience Research Reporting: The ARRIVE Guidelines for Reporting Animal Research. *Plos Biology* 8(6):e1000412; doi: 10.1371/journal.pbio.1000412.
34. Gladden JE, Smock LA. 1990 Macroinvertebrate distribution and production on the floodplains of two lowland headwater streams. *Freshwater Biology* 24:533–545; doi:10.1111/j.1365-2427.1990.tb00730.x.
35. Chapman JM, Proulx CL, Veilleux MAN, Levert C, Bliss S, André ME, et al. 2014 Clear as mud: A meta-analysis on the effects of sedimentation on freshwater fish and the effectiveness of sediment-control measures. *Water Research* 56:190–202; doi:10.1016/j.watres.2014.02.047.
36. Feld CK, Fernandes MR, Ferreira MT, Hering D, Ormerod SJ, Venohr M, et al. 2018 Evaluating riparian solutions to multiple stressor problems in river ecosystems - A conceptual study. *Water Research* 139:381–394; doi:10.1016/j.watres.2018.04.014.
37. Schenk ER, McCargo JW, Moulin B, Hupp CR, Richter JM. 2015 The influence of logjams on Largemouth Bass (*Micropterus Salmoides*) concentrations on the lower Roanoke River, a large sand-bed river. *River Research and Applications* 31:704–711; doi:10.1002/rra.2779.
38. Müller S, Wilson C, Ouro P, Cable J. submitted Experimental investigation of physical leaky barrier design implications on juvenile rainbow trout (*Oncorhynchus mykiss*) movement.
39. Hurst Jr PM. 1953 Can fish see color?. *The progressive fish-culturist* 15(2):95–95; doi:10.1577/1548-8640(1953)15[95:CFSC]2.0.CO;2.
40. Bardach JE. 1950 Do fish have color vision?. *Bios* 21(4):273–275.
41. Kawamura G, Matsushita T, Nishitai M, Matsuoka T. 1996 Blue and green fish aggregation devices are more attractive to fish. *Fisheries Research* 28:99–108; doi:10.1016/0165-7836(96)00478-X.
42. Nakano N, Kawabe R, Yamashita N, Hiraishi T, Yamamoto K, Nashimoto K. 2012 Color vision, spectral sensitivity, accommodation, and visual acuity in

1
2
3
4
5
6
7
8
9
10
11
12
13
14
15
16
17
18
19
20
21
22
23
24
25
26
27
28
29
30
31
32
33
34
35
36
37
38
39
40
41
42
43
44
45
46
47
48
49
50
51
52
53
54
55
56
57
58
59
60

juvenile masu salmon *Oncorhynchus masou masou*. *Fisheries Science* 72:239–249; doi:10.1111/j.1444-2906.2006.01144.x.

43. Heggenes J, Traaen T. 1988 Daylight responses to overhead cover in stream channels for fry of four salmonid species. *Holarctic Ecology* 11:194–201.

44. Harris JM, Paukert CP, Bush SC, Allen MJ, Siepkner MJ. 2017 Diel habitat selection of largemouth bass following woody structure installation in Tabel Rock Lake, Missouri. *Fisheries Management and Ecology* 25(2):107–115; doi:10.1111/fme.12266.

Appendix B

RSOS-201843: Leaky barriers: leaky enough for fish to pass? Response to reviewers' comments

28th January 2021

Associate Editor
Dear Dr Mark Smith,

Thank you for the reviews of our manuscript and valuable suggestions from the two reviewers.

Based on the Reviewer's valuable suggestions, we have refocused the original manuscript to focus on the fish movement in the vicinity of a porous and non-porous channel-spanning structure which may be associated with a simplified representation of a leaky barrier. Additional information on leaky barrier structures found in the field are added, showing the relation between the investigated simplified and scaled structures and the ones found in the field.

In addition to the submitted version, we explain below how we have addressed the reviewers' comments. We have outlined the reviewers' comments in **bold font** and our responses describing amendments made to the manuscript in regular font. Line numbers provided in our response refer to the amended text in blue in the track changes version.

Yours sincerely,

Stephanie Mueller, Dr Catherine Wilson, Dr Pablo Ouro and Prof Jo Cable

1. Editor

Summary:

Both reviewers agree in their suggestions of major reviews and present several helpful suggestions. Scaling issues need to be addressed and clarified as should the relationship between the experiments conducted and the passage of fish through leaky barriers in the real world. This may require a re-focusing of the paper and some additional experimental details, which constitute a major revision.

We are very grateful for the comments provided and the opportunity to submit a revised version of the manuscript. The manuscript has been refocused toward the investigation of an idealised porous and non-porous barrier, derived from leaky barriers found in the field. We have also clarified the relationship between our model barriers and those present in the field.

2. Reviewer 1

2.1 Summary:

The experiments are interesting and it is challenging to perform hydraulic experiments with living organisms. I do think there is a publishable study in here but, in its current form I have concerns detailed below.

2.2 General comments:

The main one is that, to me at least, these experiments are highly abstract and not comparable to conditions in ‘field scale’ rivers – the scaling in these experiments does not appear to be meaningful for the fish, leaky dam or the hydraulics. Therefore, I have concerns about the relevance of the findings to leaky dams and fish in rivers. However, I think this is made into a bigger issue than it needs to be due to the pitching of the whole paper around leaky barriers. I would personally rework to focus more around fish behaviour in experimental conditions and how fish navigate around and through structures, and then have a much more brief mention at the end of the discussion that these results, with further work, may be of relevance for x,y,z reasons, one of which may very well be leaky dam design. I hope the comments below help the authors.

Many thanks for providing so many valuable comments. The design of the river and leaky barrier model was based on the geometric scaling using four length scales which characterise the physical properties of the stream and leaky barriers at Wilde Brook, Corvedale (Shropshire, UK). The model to prototype scale was approximately 1:7 (1:6.7) and based on the (i) channel width; (ii) the bankfull depth; (iii) the vertical gap underneath a leaky barrier and; (iv) log diameter (more details are given below in 2.3.4). For this model scale, the juvenile salmon length scale (e.g. standard length) used in these experiments (average $L_{\text{model}} = 93.3$ mm) correspond to the length of adult salmon ($L_{\text{field}} = 653$ mm; annual mean range is $500 \leq L_{\text{field}} \leq 800$ mm in a study by Bacon et al. 2009) so again this aspect complies with geometric similarity. In terms of the kinematics, the bulk velocities and flowrates selected for the experiments are in the correct range to comply with Froude similarity (further explanation is given below in 2.3.4).

Details of the geometric scaling, and kinematic scaling (Froude law) are now included in the revised manuscript (lines 173-186 and lines 148-156 respectively).

Furthermore, we have also followed the Reviewer's advice and changed the pitch of the paper to fish movement through and around instream hydraulic structures by re-writing the introduction and editing the discussion (lines 20-120 and lines 494-504, respectively).

2.3 Detailed comments:

2.3.1 Page 1, Line 10 beginning "If global warming..." – should the reference be at the end of the sentence? Presumably the 220% is a prediction by the paper referenced earlier in this sentence.

This comment is now redundant as this sentence, and paragraphs 1 and 2 have now been replaced with new paragraphs that focus on fish movement through and around instream hydraulic structures, to address 2.2 above.

2.3.2 Page 2, Line 25: there is a typo in this sentence.

Thank you, typo has been amended.

2.3.3 Page 2, Line 38 beginning "The direction..." reads strangely to me.

Sentences has been reworded (lines 126-129).

2.3.4 Page 2, Line 46 - 50: I found this section a bit confusing. I understand it is a scale model but scaled to what? – I am guessing this isn't a Reynolds or Froude scaling approach but, some information on how the flow is scaled relative to barriers is needed if results are going to be used to inform field-scale work.

In addition, are these flows scaled to a particular river? The discharge seems low relative to the depth, meaning the velocity is high (presumably around 0.3 m/s) – I cannot see how these conditions could be 'scaled up' to be relevant to field-scale river conditions.

A brief explanation is given above in response to 2.2.

The design of the river and leaky barrier model was based on the geometric scaling of four length scales which characterise the physical properties of the stream and leaky barriers at Wilde Brook, Corvedale (Shropshire, UK). The model to prototype scale was approximately 1:7 (1:6.7) and based on geometric scaling of the (i) channel width; (ii) bankfull depth; (iii) vertical gap underneath a leaky barrier and; (iv) log diameter. For Wilde Brook the channel's b_{mc}/h_{mc} ratio varies in the range $1.66 \leq b_{mc}/h_{mc} \leq 4.8$, based on 10 selected cross-sections and a set of 105 observations. The b_{mc}/h_{mc} ratio of 4 was chosen for this study and previous studies on bed erosion (Follett and Wilson, 2020) as this typifies the channel and this ratio was maintained in our lab model. At Wilde Brook the leaky barriers have a vertical gap (b_0) to bankfull height ratio (b_0/h_{mc}) in the range of $0.333 \leq b_0/h_{mc} \leq 0.5$, which is typical of many leaky barriers in the field and a b_0/h_{mc} of 0.333 was maintained for our laboratory model.

Geometric scaling was applied for both barrier model designs using a dowel diameter, d_{model} , of 25 mm, which represents a typical field log diameter in the range $0.17 \leq d_{field} \leq 0.33$ m which is in keeping with the leaky barriers at Wilde Brook and other sites of this scale.

The bulk velocities and flowrates selected for the experiments are in the correct range to comply with Froude similarity. For Froude similarity, velocity and discharge scale using the following relationships: $U_{field} = U_{lab}\sqrt{\lambda}$ and $Q_{field} = Q_{lab}\lambda^{5/2}$ respectively where $\lambda = 6.7$. At Wilde Brook we do not have a measurement of discharge at the leaky barrier locations and the selected lab discharge correspond to field scale discharges of 2.55 and 3.25 m³s⁻¹, for the 0.8Q_{bf} and Q_{bf} conditions respectively, which is in keeping with the field channel scale (bankfull flow area = 4 m²). The lab bulk velocity of 0.32 ms⁻¹ for bankfull conditions corresponds to a field scale velocity of 0.85 ms⁻¹, which seems a reasonable magnitude for a stream flowing at full bankfull capacity before inundating the floodplains. The text has been amended to include some of this information (lines 148-156; lines 173-186).

2.3.5 I also wanted to see a breakdown of control flow conditions at this point but I noticed it came later. Might be worth referring to that table here too.

A sentence referring to the control flow conditions in the table is now added to section 2.1 (lines 139-140).

2.3.6 Page 2, Line 54: Given the condition is 80% bankfull it might be best to describe it as a “higher return period” than as a “higher flood return periods”

It has been clarified in the text that 80% bankfull flow events have a higher probability of occurrence in one year than a larger magnitude 100% bankfull event (lines 143-145).

2.3.7 Page 3 – Figure 1 is useful but I still have difficulty visualising the barriers. A photograph of the barriers would be a useful addition.

A photograph of the porous and non-porous leaky barriers used in the experiments has been included in the amended introduction, along with two leaky barrier photographs showing a porous and non-porous barrier found in the field (new Figure 1).

2.3.8 Page 3, Line 30 - 44: “scaled from those installed in the field”. As above, more information about this scaling is needed – what measurements of the field barriers were made and how was the scaling done? For example, does the 12.5 mm gap scale to any field measure?

The manuscript states that two leaky barrier porosities were analysed but no information is given on where these come from and whether they are based on field measurements. I suspect these are completely abstract structures which is not a problem in itself but, I do think a refocusing of the manuscript away from ‘recreating leaky barriers in rivers’ is probably needed.

Please refer to 2.3.4 above for further information on the corresponding field site and scaling, and this information is included in lines 173 to 186 of section 2.2.

Information on porosity is provided in sections 1 (lines 75-77) and 2.2. (lines 166-168), which explains how the non-porous barrier represents a barrier whereby the natural accumulation of brush material and sediment between the log members occurs over time. The porous barrier, on the other hand, represents the original state of the barrier after constructions, allowing flow through.

The introduction has now been rewritten and refocused on fish movement around an instream hydraulic structures and how they can act as physical barriers and “velocity” barriers to fish movement. Furthermore, it is explained in 2.2 and 2.3.4 above that the physical dimensions of the lab channel and leaky barrier was based on a field site at Wilde Brook (Shropshire) and therefore porosity confirms with geometric similarity, given that it is a function of the log diameter, log orientation and barrier width/length/depth. Photographs of the leaky barriers from the lab and the field are now included in Figure 1.

2.3.9 Page 5, line 51: Maybe I have misunderstood but the water depth is 15 cm yet the fish are approximately 12 cm long. Again, the scaling here seems way off to me – especially if this is meant to be bankfull flow. Given it is known the ability of salmon to overcome and navigate barriers is dependent on the ration of upstream and downstream water depth this strikes me as a problem. Plus, the ‘gap’ under the barrier was 5 cm high which again, doesn’t strike me as a relevant fish-structure scaling for a real world situation. As above, this is much more of a problem if the purpose is to recreate river conditions, which appears to be the aim. It is much less of a problem if the focus is on fish behaviour more generally around porous and non-porous structures.

For the model scale, the juvenile salmon length scale (e.g. standard length) used in these experiments ($L_{\text{model}} = 93.3 \text{ mm}$) correspond to the length of adult salmon ($L_{\text{field}} = 653 \text{ mm}$) so this aspect complies with geometric similarity. The flow depth also complies with geometric similarity as explained above in 2.2 and 2.3.4. However, while a model vertical gap (b_0) of 50 mm scales to field conditions (the ratio b_0/H and its scaling is covered in 2.3.4) and the channel/leaky barrier scale at Wilde Brook, it is acknowledged that fish to barrier scaling in our model is unrealistic. It only works in field conditions where the vertical gap would be large enough for adult salmon reaching head waters to spawn and juvenile salmon as used in our experiment. This point is now made in the text (lines 499-500).

The focus has now been changed to fish movement around and through a porous and non-porous barrier.

2.3.10 Page 7, line 54: I think it would be useful to define the 3 regions, particularly “under flow” – presumably this means the flow beneath the structure in the opening but confirmation of this would be good.

“beneath the structure” has been added to the sentence to clarify (line 326).

2.3.11 Page 7, line 55 beginning “In all cases...” this sentence is worded oddly to me.

Amended, thank you.

2.3.12 Page 8, line 53: typo and strange characters in this sentence.

Amended, thank you.

2.3.13 Page 10 – Figure 5 – I got really confused here but, I think the colours are mislabelled in the caption. Downstream is dark blue not green, etc. Also, should figure 5c be colour coded?

Yes, figure caption has been corrected, thank you (Figure 6).

2.3.14 Page 11 line 38 – 45: I agree with this summary of the findings, which are interesting in informing about how fish interact with an abstract obstacle but I just do not see given the scaling of fish to barrier, and the depth to velocity scaling of the flow, how this can really be related to fish behaviour around field-scale, leaky barriers.

We agree the scaling of the fish to the vertical gap of the barrier is problematic and this is now acknowledged in the text (lines 499-500). However, the model does conform to geometric similarity and Froude similarity and this is covered above 2.2 and 2.3.4). The introduction has now been rewritten which changes the focus the paper, see above 2.2 (lines 20-120).

2.3.15 Page 12, line 27: What colour was the polythene wrapping?

Information has been added within the text (line 463). The structure was wrapped in orange polythene.

2.3.16 Page 12, line 35: I don't find the colour argument that convincing. Through rheotaxis the fish will know which way is upstream. I do agree that the colour may influence their behaviour.

Number of passes has been used as measure for the fish's activity, which was found to increase using the orange wrapped barrier as oppose to the natural looking porous barrier independent of discharge. Colour may present a visual cue and or attraction factor. Further experiments will be needed to identify causal relationships. Colour is only one possible influencing factor, but the complexity of fish behaviour has been addressed in the discussion section, so we hope you agree this point is now addressed (lines 496-498).

2.3.17 Page 12, line 40: There is a transition here form discussion about colour to overhanging logs. I would separate into 2 separate paragraphs

Paragraph has been separated into two parts.

2.3.18 Page 12, line 46-57: I agree but none of this content is really informed by the experimental findings.

We disagree, as it has clearly been shown that fish could move between the downstream and upstream part of the test section provided the presence of a vertical gap beneath the barrier. This has been indirectly quantified by the mean number of upstream passing fish and mean number of passes per fish. Management of the barriers is important to ensure the regular maintenance of this gap to ensure longitudinal river connectivity.

2.3.19 Page 13, line 8 – 12: I agree with all this and I think this is an interesting finding. The second half when referring to leaky barriers I find more problematic because there are so many confounding influences and because the results here are far from conclusive.

The second half of this paragraph has now been amended to reflect the new focus of the paper (line 511-517) and we clearly state that further research will be needed for results to be projected to leaky barriers in the field.

3. Reviewer 2

3.1 Summary:

This paper presents an experimental study of flow fields and fish behaviour around leaky wooden dams of two different porosities and at two different discharges. Leaky wooden dams are increasingly being used in small headwater streams to contribute to "slowing the flow".

The authors are very grateful for the valuable comments discussing scaling and real-world application issues as well as the additional comments provided in the manuscript. Please find below our responses to the individual comments.

3.2 General comments:

3.2.1 scaling of channel and wood dimensions (scaling is not adequately presented; are you Froude scaling or merely applying geometric similitude? We need more information about the field prototype to know)

The focus of the paper has now been shifted away from exactly replicating leaky barriers and flow conditions found in the field.

Additional information, on where the design of the porous and non-porous structure came from has been added in section 2.2. (lines 173-186), including information about the present structures found in Shropshire and an approximate scale.

The design of the river and leaky barrier model was based on the geometric scaling using four length scales which characterise the physical properties of the stream and leaky barriers at Wilde Brook, Corvedale (Shropshire, UK). The model to prototype scale was approximately 1:7 (1:6.7) and based on geometric scaling of the (i) channel width; (ii) bankfull depth; (ii) vertical gap underneath a leaky barrier and; (iii) log diameter. For Wilde Brook the channel's b_{mc}/h_{mc} ratio varies in the range $1.66 \leq b_{mc}/h_{mc} \leq 4.88$, based on 10 selected cross-sections and a set of 105 observations. The b_{mc}/h_{mc} ratio of 4 was chosen for this study and previous studies on bed erosion (Follett and Wilson, 2020) as this typifies the channel and this ratio was maintained in our lab model. At Wilde Brook the leaky barriers have a vertical gap (b_0) to bankfull height ratio (b_0/H) in the range of $0.333 \leq b_0/h_{mc} \leq 0.5$, which is typical of many leaky barriers in the field and a b_0/h_{mc} of 0.333 was maintained for our laboratory model.

Geometric scaling was applied for both barrier model designs using a dowel diameter, d_{model} , of 25 mm, which represents a typical field log diameter in the range $0.17 \leq d_{field} \leq 0.33$ m which is in keeping with the leaky barriers at Wilde Brook and other sites of this scale.

The bulk velocities and flowrates selected for the experiments are in the correct range to comply with Froude similarity. For Froude similarity, velocity and discharge scale using the following relationships: $U_{field} = U_{lab} \sqrt{\lambda}$ and $Q_{field} = Q_{lab} \lambda^{5/2}$ respectively where $\lambda = 6.7$. At Wilde Brook we do not have a measurement of discharge at the leaky barrier locations and the selected lab discharge correspond to field scale discharges of 2.55 and 3.25 $m^3 s^{-1}$, for the 80% Q_{bf} and 100% Q_{bf} conditions respectively, which is in keeping with the field channel scale (bankfull flow area = 4 m^2). The lab bulk velocity of 0.33 ms^{-1} for bankfull conditions

corresponds to a field scale velocity of 0.85 ms^{-1} , which seems a reasonable magnitude for a stream flowing at full bankfull capacity before inundating the floodplains. The text has been amended to include some of this information (lines 148-156).

3.2.2 scaling of fish relative to the above (how does the width and height of your model salmon compare to prototype salmon? How does this scale compare to your channel scale (width, depth)? How do either of these scales compare against your wood structure scale (both the elements and their spacing)? How do all these compare against the scale of the burst and/or sustained swim speed of your model salmon to prototype salmon?)

For this model scale, the juvenile salmon length scale (e.g. standard length) used in these experiments (average $L_{\text{model}} = 93.3 \text{ mm}$) correspond to the length of adult salmon ($L_{\text{field}} = 653 \text{ mm}$; annual mean range is $500 \leq L_{\text{field}} \leq 800 \text{ mm}$ in a study by Bacon et al. 2009) so this aspect complies with geometric similarity (lines 231-233). The flow depth also complies with geometric similarity as explained above in 3.2.1. However, while a model vertical gap (b_0) of 50 mm scales to field conditions and the channel/leaky barrier scale at Wilde Brook, it is acknowledged that fish to barrier scaling in our model is unrealistic. It only works in field conditions where the vertical gap would be large enough for adult salmon reaching head waters to spawn and juvenile salmon as used in our experiment. This point has now been made in the text (lines 499-500).

Swimming performance tests conducted by Palstra et al. (2020) determined a critical swimming speed of approx. 0.9 m/s for fish of standard length 123mm, which is a similar size fish compared to our model (standard length 93.3mm). As mean streamwise velocities greatly vary within the test section, fish were exposed to a wide range of velocities, ranging from approx. 0.2 to 0.7 m/s, which lies within the fishes sustain swimming speed. Scaling of burst/sustained swimming speed of our model fish to model barrier and field condition, however, is beyond the scope of this study.

3.2.3 fish behaviour- generally speaking, a juvenile salmon doesn't really want to go up or downstream and it would be perfectly fine/happy if it never passed your barrier. If it was an anadromous salmonid (the next stage in the life cycle), it would swim seaward when the time was right, but given their size I don't think that would happen for another year. So the question is the extent to which "motivation" was dominated by the fright response of the fish to being prodded and whether fish were hiding underneath the porous barrier and exhibiting avoidance behaviour when they swam upstream? Unfortunately, we cannot tell. I think that all prodded fish must be removed from your analyses in order to eliminate these possibilities.

Due to the shift of focus, the paper now discusses the free fish movement near a porous and non-porous structure, not necessarily linking their movement to fish migration. The number of passes has solely been used as measure for the fish's activity and spatial preference as well as to demonstrate that the provided vertical gap presents an essential feature of the structures design ensuring longitudinal river and habitat connectivity. At the time experiments were conducted, fish at the parr-early smolt stage, so we have now made it clear in the amended manuscript (lines 238-240) of the migratory phase of these fish. Importantly though, all these fish would still be moving between habitats to seek shelter, forage and rest in low velocity regions.

Here we investigate fish movement in response to a porous and non-porous structure. Their behaviour is likely to be influenced by flow alterations caused by the different structures, barrier colour or as highlighted by the reviewer the gentle nudging of inactive fish during the experiment. In our study, only fish furthest downstream, not showing any behaviour were subject to this human interaction. Repeated nudging without a response resulted in the removal of these fish from the data analysis. To account for any potential effect of human interference, we analysed the data twice, including (N=77 fish) and excluding those fish subject to human interference (N=50 fish). There was no significant difference in the independent variables affecting the percentage of upstream passing fish, floodplain usage and time spent upstream, downstream and beneath the barrier. Only the dependent variable “passes per fish” resulted in a different result if prodded fish were excluded, showing that neither flow condition (GLM, $p=0.8893$) nor barrier (GLM, $p=0.119$) had a significant impact while barrier was found to be significant factor influencing fish behaviour (GLM, $p=0.0174$) when prodded fish remained included. We have clarified this now in the results of the paper and added in this additional information (line 409-415) which we agree greatly improves the manuscript.

3.2.4 the real-world applicability of leaky wooden dams preventing access to headwaters by adult salmon; leaky wooden dams are generally installed in headwater streams that are not reachable by adult salmon, so it is unclear how applicable your results are in the real world.

In addition to the above, i think that this study would benefit from inclusion of additional velocity data and exploration of the jet recovery downstream of structures. I have seen this data presented in non peer-reviewed outlets by your team, so it should not be too challenging to include it here; i think that the contrast between hydraulics of porous/non-porous barriers is of significant interest.

Although most rivers in the UK are fragmented, great effort is being made to remove or retrofit barriers to allow unhindered fish migration. The authors agree that at the current stage not all rivers equipped with leaky barriers may be reachable by adult salmon. However, other migratory and non-migratory species may be present, and we aim here to demonstrate proof of concept.

Due to the greater impact of the physical design of the barrier, less focus was given here to the channel hydrodynamics. A more detailed study analysing upstream and downstream hydrodynamics would change the focus of the paper and disperse its message.

3.3 Detailed comments:

3.3.1 Page 3 Have you got photos and or dimensions of the field prototype?

Added to the amended manuscript (new Figure 1).

3.3.2 Page 3 So scaled wood but not scaled fish?

As demonstrated in 3.2.1 and 3.2.2, both channel/barrier and fish scale comply with geometric similarity. For this model scale, the juvenile salmon length scale (e.g. standard length) used in these experiments (average $L_{\text{model}} = 93.3$ mm) correspond to the length of adult salmon ($L_{\text{field}} = 653$ mm; annual mean range is $500 \leq L_{\text{field}} \leq 800$ mm in a study by Bacon et al, 2009) so this aspect complies with geometric similarity.

3.3.3 Page 3 Line 38/39 reorder/rephrase

Amended, thank you.

3.3.4 Page 3 Fish of length 93mm have a width of about 10mm. Channel width is 600mm, so channels are 60 x width of a fish. For an adult salmon, width ~15-20cm, so scaled-up channel width (relative to the fish) would be ~9-12m. Are leaky dams fitted to rivers as wide as this?

The scaling of the leaky barriers, channel and fish is covered in 3.2.1 and 3.2.2. Sections of the original manuscript have been amended (lines 173-186; lines 231-233).

3.3.5 Page 3 Line 48 No slope adjustment required?

No slope adjustment was required to establish subcritical flow conditions for both discharges. Uniform flow conditions, where the energy slope is parallel to the water surface slope and bed slope, was established prior to the installation of the barriers and this is fully explained in section 2 (a) (line 130, lines 145-147). Installation of the barriers resulted in a change in water surface profile for each flow condition and this information is presented in Table 1, section 2.2.

3.3.6 Page 3 Line 54 A bankfull event has a return period of, say 2.33 years. An 80% bankfull event would have a return period less than this because it occurs more frequently? This should either be "shorter flood return period" or "higher flood exceedance frequency"

It has been clarified in the text that an 80% bankfull event has a higher probability of occurrence in one year than a larger magnitude 100% bankfull event (lines 143-145).

3.3.7 Page 4 Line 30 How? We need more information about the prototype. Your group gives some of this in Follett and Wilson (2020), RiverFlow2020, pp.735-742 but i think we need to know more- how do the dowel dimensions scale to the real world? How do the gaps scale to the real world?

The focus of the amended manuscript has changed based on the suggestion by Reviewer 1, investigating fish movement and behaviour around a porous and non-porous hydraulic structure which approximates a leaky barrier design. Therefore, new paragraphs one and two of the introduction now address the impact of instream hydraulic structures on flow alterations and fish movement (line 20-54). Additional details on the design choices and scale have been added to section 2.2 (line 173-186) and are discussed in 3.2.1 above.

3.3.8 Page 5 Line 39 How many points were removed (number and percentage)? How were removed points treated? Were they merely ignored in the statistical analysis (i.e. left as NaNs) or interpolated. If the latter, which interpolation method was used?

In a pre-filtering step, timesteps with insufficient SNR or Correlation (<15dB and <70%, respectively) were removed from the dataset. Detailed information on the number of points removed for each measurement point were not recorded. These points were deleted from the time series.

3.3.9 Page 5 Line 40 It appears that the method of Mori et al attempts to replicate the method of Wahl (2003), except that the universal threshold proposed by Goring and Nikora (2002) is retained, rather than the Chauvenet criterion used by Wahl. This distinction is important, and results in fewer points being filtered. Irrespective, i would like the authors to write "using the despiking filter of Mori et al. (2007), which is a modification of the Wahl (2003) 3D phase space threshold filter"

We agree and have amended the sentences accordingly.

3.3.10 Page 5 Line 46 Which capture mode did you use? Did you have to remove lens distortion? If so, how?

Capture mode and resolution used have now been added into the description (line 255). As videos were only used as a visual aid and not to extract details from it, there was no need to consider lens distortion.

3.3.11 Page 5 Line 58 How did you perform these tests? Are these burst swim speeds (over 20 seconds) or sustained swim speeds? How many tests did you perform?

This seems to be a misunderstanding. The reference at the end of this sentences indicated this study was performed by Palstra et al. (2020) and not by the authors. The information was added to show the physical swimming capability of the fish chosen. The text has now been amended to make this clearer (lines 240-243).

3.3.12 Page 6 Line 23 How does swimming performance scale relative to how you have scaled the structures?

Please see our reply to comment 2.3.9. Swimming performance has not been measured by the authors and is beyond the scope of this paper.

3.3.13 Page 6 Line 44 So you are expecting juvenile salmon, which either have no inclination to move or a inclination to swim/drift downstream, to swim upstream? And if they don't (35 out of 86 cases), you attempt to prompt a flight behaviour, resulting in 26 out of the remaining 77 having been prompted?

No, we did not assume fish would swim upstream, but we expected them to show a range of natural behaviours (e.g. station holding, active swimming, drifting).

Please see point 3.2.3 which discusses the implications of including all fish into the statistical analysis.

3.3.14 Page 8 I think that this would benefit from some of the data, description and explanation that you provided in Follett et al. (https://www.therrc.co.uk/sites/default/files/files/Conference/2019/Posters/25_follett_elizabeth.pdf), in particular the exploration of jets and flow recovery

This paper focuses on the fish response to porous and non-porous barrier and the aim to determine whether discharge or physical design has greater importance. Detailed hydrodynamic results are outside the scope of this study.

3.3.15 Page 9 Line 53 Not if you adopt your chosen significant p-value

Thank you, amended in the text.

3.3.16 Page 10 Figure Does not match the figure caption! Legend does not match the plot. Rename "data1" Make clear which column is which discharge case Does not match the legend! Does not match figure title! I don't think i'm colour blind and i see this as grey? Error bars for control and porous cases appear to be larger than the mean number of passes per fish Page 11 Line 7 grey?

Thank you, graph, legend and figure caption have been corrected. Colour was maintained because of the 10 people we asked to view this – they all could easily distinguish the colours.

3.3.17 Page 11 Line 13 $p > 0.05$

Exact p-value has been maintained throughout the results section to provide higher accuracy. Only results with a p-value much smaller than 0.001 were noted as $p < 0.001$.

3.3.18 Page 10 Figure Error bars for control and porous cases appear to be larger than the mean number of passes per fish

That is correct, error bars represent the standard deviation of the number of passes per fish. Each was allowed to pass multiple time, for example for the 80% control case fish passed between 0-5 times. The large standard deviation, therefore, indicates a greater variation of passes per fish.

3.3.19 Page 11 Line 18-25 I'm sorry but i just don't see how the mean number of passes can be significantly greater than zero for either the no barrier or the porous barrier cases

Please see reply to previous comment. The majority of fish passed at least once from downstream to upstream. However, some fish passed multiple times, increasing the mean number of passes per fish.

3.3.20 Page 11 Line 58 Recall comment about scaled channel width and stage of salmon life cycle

Thank you, amended in the text.

3.3.21 Page 12 Line 18 or see the other proposed reasons in Goodwin et al (2014; <https://doi.org/10.1073/pnas.1311874111>) and references therein

Many thanks for pointing out this very interesting paper. Suggested further reasons have now been incorporated into the text.

3.3.22 Grammatical/language comments provided in the manuscript.

Many thanks for providing additional comments in the manuscript. All changes have been accepted.

4. Notations

b_0	vertical gap beneath barrier structure
b_{mc}	main channel width
d	dowel/log diameter
h_{mc}	main channel height
L	standard length
Q	discharge
$100\% Q_{bf}$	100% bankfull discharge
$80\% Q_{bf}$	80% bankfull discharge
U	bulk velocity
λ	scaling factor

Indices:

Model	Laboratory conditions
Field	Field conditions

5. References

Bacon, P. J., Palmer, S. C. F., Maclean, J. C., Smith, G. W., Whyte, B. D. M., Gurney, W. S. C. and Youngson, A. F. (2009) empirical analysis of the length, weight and condition of adult Atlantic salmon on return to the Scottish coast between 1963 and 2006, *ICES Journal of Marine Science*, 66, 844-859.

Follett, E. M. and Wilson, C. A. M. E. (2020) Bedload transport induced by channel-spanning instream structures, Presented at: 10th Conference on Fluvial Hydraulics (RiverFlow 2020), Delft, Neatherlands, 7-10 July 2020, Published in: Uijttewaal, Wim, Franca, Mário J., Valero, Daniel, Chavarrias, Victor, Arbós, Clàudia Ylla, Schielen, Ralph and Crosato, Alessandra eds. *River Flow 2020 Proceedings of the 10th Conference on Fluvial Hydraulics (Delft, Netherlands, 7-10 July 2020)*. London, England: CRC Press/Balkema, pp. 735-742, doi:10.1201/b22619.

Palstra A. P., Kals J., Böhm T., Bastiaansen J. W. M., Komen H. (2020) Swimming performance and oxygen consumption as non-lethal indicators of production traits in Atlantic Salmon and Gilthead Seabream, *Frontiers in Physiology*, 11:759, doi: 10.3389/fphys.2020.00759.